# Aging and episodic memory specificity: Evidence challenging a domain-general pattern separation decline

Ariana Youm[1,2], Melanie Cohn[1,2‡], Katherine Duncan[1‡*]

1 Department of Psychology, University of Toronto, Toronto, Ontario, Canada, 2 Krembil Brain Institute, University Health Network, Toronto, Ontario, Canada

‡ These authors are senior authors on this work.
* katherine.duncan@utoronto.ca

## Abstract

Aging is associated with a decline in episodic memory specificity. This phenomenon has been observed across various memory tasks, such as the Mnemonic Similarity Task (MST), where older adults show a heightened tendency to falsely recognize perceptually similar items. While many studies suggest this impairment reflects a generalized reduction in pattern separation, others indicate that older adults may exhibit preserved discrimination abilities for semantic representations. Furthermore, pattern separation accounts also posit that a bias toward pattern completion, a process whereby partial cues reactivate whole representations, occurs with aging. However, the evidence for this shift remains mixed, which may be in part due to how pattern completion is commonly measured. The current study aimed to investigate whether aging affects memory discrimination for semantically similar content, using tasks that minimize reliance on visual-spatial processing and executive functioning, both of which tend to decline with age. We designed two independent tasks that respectively tax memory specificity and pattern completion: the Verbal Discrimination Task (VDT) and the Verbal Completion Task (VCT). Sixty-six younger adults and 66 older adults completed these tasks, and a subset also completed the Object MST (younger n = 33 and older n = 35) to allow for a direct comparison between visual and semantic similarity. Our results showed that, compared to younger adults, older adults exhibited greater deficits in memory specificity for perceptually similar lures (Object MST) than for semantically similar content (VDT), suggesting that age-related memory impairments may be more pronounced in perceptual domains. Additionally, older adults showed reduced performance on the VCT, suggesting that deficits in memory specificity may be independent of deficits in pattern completion. Together, these findings go against the view that age-related differences in memory specificity are strictly rooted in a modality-general pattern separation deficit.

**Data availability statement:** All data and code can be found in this OSF repository: https://osf.io/jkcg9/?view_only=95d52ace925a4e6ab-339d760e50ad351.

**Funding:** K.D. received Canada Research Chair: CRC-2019-00107; Ontario Early Researcher Award: ER18-14-139; CFI/ORF-2014-34479; NSERC: RGPIN-2023-04231. A.Y. received Canada Graduate Scholarships - Doctoral award, the General Motors Women in STEM award, and a stipend from CIHR grants awarded to M.C.: FRN 162197.

**Competing interests:** The authors have declared that no competing interests exist.

## Introduction

Not all aspects of memory are equally affected by aging: some, such as the precision of our mnemonic representations, are particularly sensitive to age-related changes [1–3]. For example, while a young adult may easily retrieve the details of a specific New Year's Eve party, an older adult might only retrieve the gist of the event, the details blurred with the many other parties they have celebrated in the past. Indeed, prior research posits that the ability to remember specific details of past experiences declines with healthy aging, accompanied by an increase in the reliance on gist memory [4–6].

Past research suggests that reductions in hippocampal pattern separation, a process whereby similar inputs are orthogonalized into distinct representations to reduce interference [7,8], contribute to age-related loss of memory specificity [9–11]. Rodent research shows that ensembles of granule cells in the dentate gyrus subregion remap to even small changes in environmental inputs and serve as a mechanism for pattern separation [12]. Importantly, this hippocampal subregion is most sensitive to the effects of advancing age [13]. Consistent with resulting pattern separation deficits, aged rodents have difficulty detecting subtle changes in object features and spatial locations [11]. Similar age-related memory deficits have been found in humans using the Mnemonic Similarity Task (MST) [14–16], in which participants must discriminate perceptually similar lures from studied targets of items such as images of common objects. Across different versions of the MST, older adults were more likely to judge similar lures as being 'old' (previously studied) than younger adults. By contrast, performance on other trial types (e.g., old targets and new unrelated distractors) did not differ significantly across groups, suggesting reduced pattern separation [15,17]. Further, this behavioural profile has been related to specific functional changes in the CA3/dentate regions of the hippocampus [9,18]. In sum, this work suggests age-related impairments in discriminating between similar experiences in memory are related to the aging hippocampi's bias away from storing distinctive memory traces.

The pattern separation account predicts reduced memory specificity across content domains; similar stimuli should be hard to discriminate in memory, regardless of the way in which they are similar. Yet, the present evidence remains mixed in studies assessing memory specificity in the semantic domain. Studies using the Deese-Roediger-McDermott paradigm, in which participants are presented with lists of semantically related words, have shown that older adults are prone to falsely recalling and recognizing semantic lures as having been on studied lists [19]. Even the commonly observed effects of aging on the object MST have been shown to depend on prior semantic knowledge of objects [20,21]. To more directly probe semantic specificity, other studies have manipulated semantic similarity in adapted versions of the MST, but with mixed results. For example, one recent study used parametrically manipulated semantic similarity of adjective-noun phrases to show that older adults exhibit heightened false recognition of moderately similar lures [22]. In contrast, however, other studies suggest that older adults may have preserved memory specificity of semantic details relative to their younger counterparts. For

example, instead of using the standard pictorial MST stimuli, Ly, Murray, and Yassa [23] used verbal stimuli to manipulate conceptual (e.g., "closet" and "drawer") and phonological (e.g., "curtain" and "certain") interference, although without parametric control. They found that older adults were selectively impaired when discriminating phonological lures from studied targets, suggesting that age-related mnemonic discrimination deficits might be principally due to perceptual interference rather than conceptual interference. In another study, Delarazan et al. [24] calibrated perceptual and narrative lures for studied television show scenes to have similarly high false alarm rates. They also found that older adults exhibit a deficit in lure discriminability for perceptual but not semantic details. Thus, whether lure discrimination of semantic material in older adults is reduced or preserved is still in debate, with evidence for preservation going against the view that age-related pattern separation deficits span across modalities.

In addition to being modality-general, the pattern separation account of age-related memory changes posits that decreased performance on tasks that require pattern separation reflects a bias toward pattern completion [9]. *Pattern completion* is a process by which specific mnemonic representations are retrieved from partial or noisy input [25–27]. This reactivation makes neural representations during the retrieval cue more similar to the representation of stored memories, an increase in representational similarity that reflects the opposing transformation to pattern separation [28,29]. However, findings of age-related differences in behavioural pattern completion are mixed. On one hand, older adults' elevated false alarm rates to lures on the MST are used as evidence for a shift toward pattern completion [9,29–32]. Researchers reason that the lure is a noisy cue that reactivates related experiences, including the similar studied target, resulting in a false alarm. Yet, in addition to overactive pattern completion, false alarms could reflect age-related deficits in executive processes and strategic retrieval, which are required to monitor and use memory appropriately [33,34]. Moreover, the MST does not explicitly manipulate pattern completion processes, as it is simply indexed as a failure of pattern separation. By operationalizing the two processes this way, it is only natural that one would occur at the expense of the other. Adding credence to the pattern separation account, though, Wynn and colleagues [35] found that older adults' tendencies to false alarm to lures of studied scene images were related to the degree to which they reinstated the eye movements that they produced while studying the similar scenes. The conclusion that these false alarms were related to pattern completion was bolstered by showing that age-related differences in eye movements during encoding could not explain older adults' false alarm bias. On the other hand, age-related decrements in spatial navigation [36] and scene recognition tasks [37,38] increase as experimenters remove or degrade cues during the test, suggesting impaired pattern completion with age. While such evidence makes it enticing to conclude that pattern completion is impaired in older adults, a complication is that what looks like poor pattern completion at retrieval could reflect the encoding of poorly separated memories. Thus, the debate in the field remains.

To address these gaps, we designed a study with three objectives in mind: 1) We aimed to assess whether aging is associated with deficits in discriminating semantically (not perceptually) similar content in memory. 2) We designed a task that primarily taxes pattern completion independent of the task that primarily taxes pattern separation, rather than assuming that deficits in behavioural pattern separation reflect pattern completion. 3) We accounted for executive functioning and used tasks that were less reliant on visual-spatial processing, since these also show age-related decline and contribute to task performance [39]. To achieve these goals, we developed two independent verbal tasks; they had identical deep encoding phases, but their retrieval phases, respectively, tax memory specificity and pattern completion. The Verbal Discrimination Task (VDT) assesses the discrimination of studied abstract nouns from new unrelated words and synonyms, and the Verbal Completion Task (VCT) assesses the completion of word stems with studied words. To parse out controlled executive processes that usually contribute to cued recall performance [40–42] we used the Process Dissociation Procedure (PDP) [43] and focused on the automatic reactivation of past experiences enabled by pattern completion. As an exploratory objective, we also included the standard object MST for comparison with the VDT for a subset of participants.

We used the pattern of performance across these tasks to gain insights into the nature of age-related changes in pattern separation and pattern completion. Replicating age-related deficits in lure discrimination in the object MST but not

the VDT would suggest that memory specificity declines more steeply in perceptual than semantic domains, going against a domain-general pattern separation account. Further, older adults' performance on the VCT will independently estimate pattern completion abilities to provide another tool for assessing whether age-related changes in memory discrimination are rooted in a bias away from pattern separation.

## Materials and methods

### Transparency and openness

In this section, we report how we determined our sample size, describe all data exclusions, manipulations, and all measures in the study. All data, analysis code, and research materials are available on OSF (https://osf.io/jkcg9/overview?view_only=0ce51f6346004b59950210c7615c2702). Data were analyzed using R, version 4.3.3 [44]. This study's design and its analyses were not pre-registered.

### Participants

We recruited 92 younger adults (18–35 years) from the University of Toronto Introduction to Psychology Participant Pool and 75 older adults (55–86 years) from the University of Toronto Adult Volunteer Pool and the greater community of Toronto, ON, Canada from March 2, 2019 to October 17, 2022. A broader age range was chosen for the older adult group to introduce more variability for analyses that are outside the scope of this manuscript. From this sample, we excluded nine younger and seven older adults who did not meet the inclusion criteria of having no history of psychiatric, neurological, or learning disorders. We also excluded 17 younger and two older adults who scored below the Shipley-II vocabulary cutoff of 24, the lower limit of the broad average score for young adults or borderline range in older adults based on normative data [45]. We deemed these individuals ineligible for the study as our experimental manipulation for the Verbal Discrimination Task (VDT) relies on participants having a good grasp of the meaning of English words. This yielded a final sample of 66 younger and 66 older adults (see Table 1). We predetermined this sample size from an *a priori* power analysis to obtain 80% power. The analysis was based on a two-tailed independent samples t-test comparing younger and older adults on the pattern separation index using an effect size of $d = 0.59$ derived from a previous study [15].

Although the two age groups were sex-matched, they notably differed in their racial and ethnic makeup (Younger adults: 42% East Asian, 20% Caucasian, 15% South Asian, 11% Middle Eastern, 5% Southeast Asian, 2% Black, 6% Other; Older Adults: 67% Caucasian, 8% South Asian, 5% Black, 5% East Asian, 2% Southeast Asian, 3% Other, 12% no response) and years of education, since most of the younger adult sample were in their first year of university. Additionally, while only 44% of younger adults reported English as being their first language, 76% of older adults' first language was English. However, both samples tended to have early experience with English. The younger adults whose first language was not English, on average, acquired English at around 5 years of age ($M = 4.8$, $SD = 2.16$); likewise, the older adults whose first language was not English, on average, acquired English at around 6 years of age ($M = 5.7$, $SD = 0.95$). Raw Shipley-II vocabulary scores significantly differed across age groups ($t(127) = 10.59$, 95%CI [4.77, 6.96], $p < .0001$), but

**Table 1. Demographics of Sample Included in Current Analyses.**

|  | Group | Sex (F:M) | Years of Education | Mean Age(SD)[a] | Age Range | Mean Raw Shipley(SD) | Mean Standardized Shipley(SD) |
|---|---|---|---|---|---|---|---|
| Total Sample | Younger | 46:20 | 13.24(1.29) | 19.15(2.21) | 17-31 | 29.23(3.41) | 106.92(8.94) |
|  | Older | 46:20 | 18.49(5.54) | 72.98(8.36) | 55-86 | 35.09(2.93) | 108.29(7.27) |
| With MST | Younger | 24:9 | 13.09(1.53) | 19.24(2.82) | 17-31 | 29.00(3.26) | 106.27(8.71) |
|  | Older | 23:12 | 18.90(5.77) | 75.74(6.67) | 58-86 | 34.86(3.26) | 106.45(8.50) |

[a]SD = standard deviation.

once scores were standardized using age-matched normative data, the two groups did not significantly differ ($t(127) = 0.96$, 95%CI [−1.44, 4.17], $p = .34$; see Table 1). This matching suggests that our strict vocabulary cutoff ensured that both groups had similarly strong proficiency in English for their ages despite differences in language backgrounds. We ran exploratory analyses restricted to participants who reported English as being their first language to assess if they showed a similar directional effect as the full sample (see Results below and Supplementary Note 1). While language background minimally affected the VCT, it may have influenced the VDT (see Discussion).

We compensated younger adults with course credits for their participation and older adults monetarily for their time. All participants gave their informed written consent before the administration of the tasks, and all methods were approved by the University of Toronto Social Sciences, Humanities, and Education research ethics board.

## Experiment overview

All eligible participants completed the VCT and VDT, and half also completed the MST (see Fig 1A). After the experimental tasks, we administered a neuropsychological test battery to older adults only; these data are beyond the focus of this paper and will be reported in a separate manuscript (see Supplementary Methods). We designed the VCT, a modified

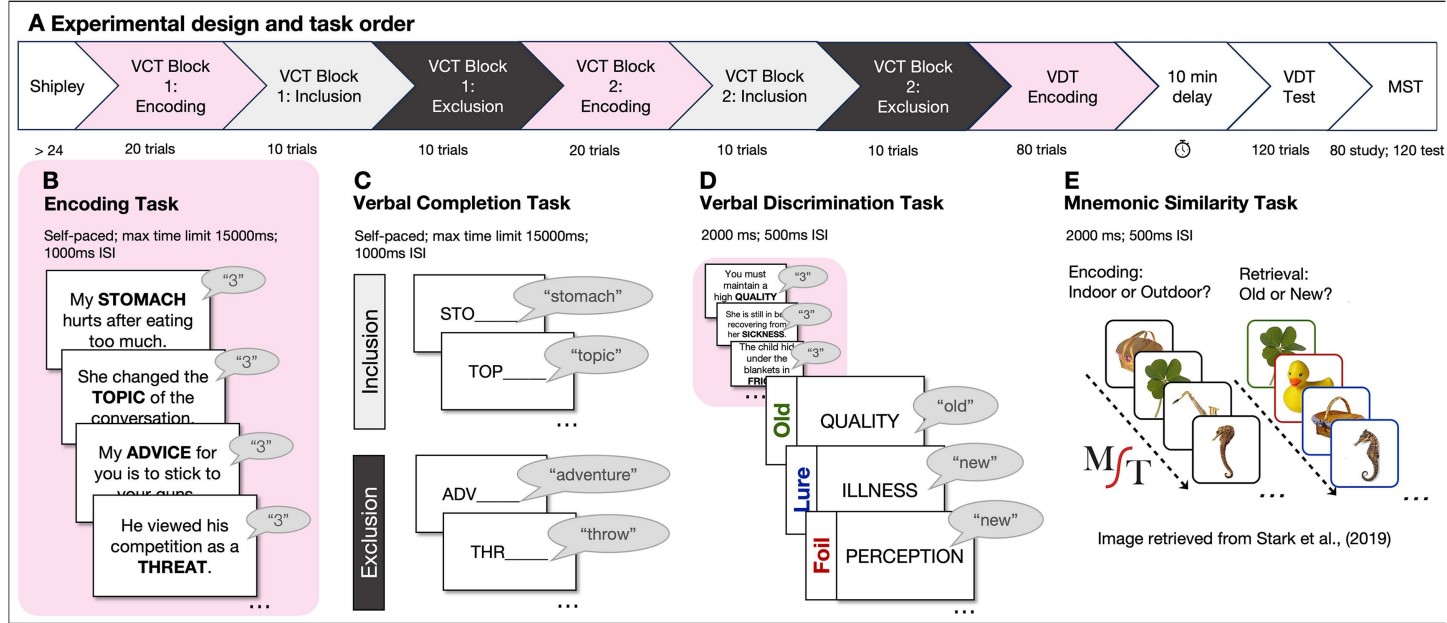

**Fig 1. Experimental Design and Procedure. (A)** Participants with scores ≥24 on the Shipley underwent all experimental tasks in the following fixed order: Verbal Completion Task (VCT), Verbal Discrimination Task (VDT), and Mnemonic Similarity Task (MST). **(B)** At encoding, participants were presented with a series of sentences that each contained one capitalized target word. Participants read each sentence out loud and then rated how well they knew the target word (1 = "don't know," 2 = "kind of know," and 3 = "know very well"). The structure of the encoding task was the same for the VCT and VDT, but the number of trials differed. During the VCT, participants were presented with 24 sentences (20 target, 2 beginning buffers, and 2 ending buffers), and during the VDT, with 84 sentences (80 target, 2 beginning buffers, and 2 ending buffers). **(C)** VCT retrieval consisted of two blocks: Inclusion and Exclusion. For Inclusion, participants were instructed to complete the cue with a word they had just studied. For Exclusion, participants were instructed to complete the cue with a word they had never seen before in the context of the experiment. **(D)** During VDT retrieval, participants engaged in a recognition memory test in which they identified each item as "old" or "new" with a button press. Forty words were exact repetitions of words presented in the encoding phase (targets); 40 were synonyms of those seen during the encoding phase but not identical (lures); 40 were new words not previously seen (foils). **(E)** 33/66 younger and 35/66 older adults completed the MST (old/new version; [15]). During encoding, participants judged whether each object was commonly found indoors or outdoors. At retrieval, participants engaged in a recognition memory test in which they identified each item as "old" or "new" with a button press. 40 were exact repetitions of images presented in the encoding phase (targets); 40 were similar images to those seen during the encoding phase but not identical (lures); 40 were new images not previously seen (foils).

stem-completion explicit memory task, and the VDT, a mnemonic discrimination task, to tax pattern completion and pattern separation, respectively. These two tasks share the same encoding task structure but have different types of test formats at retrieval. By matching the stimuli and encoding tasks, we ensured that differences in performance between tasks were due to our manipulation at retrieval, which aimed to determine the degree to which pattern separation and pattern completion were engaged. We chose to use verbal material instead of the typical visual stimuli because aging is associated with strengthened vocabulary (for review, see [46]) and reduced visuo-perceptual skills [47,48]. Thus, this approach minimizes the influence of low-level visual differences (e.g., visual acuity) across age groups, allowing us to better isolate age-related changes in representational fidelity. While memory and perception are closely intertwined processes [49,50], this design reduces the likelihood that performance differences are solely attributable to low level perceptual factors. Furthermore, we used a Process Dissociation Procedure (PDP) [43] to index pattern completion while accounting for controlled processes that tax frontal lobe function. The VCT always preceded the VDT to reduce interference during the VCT. Midway through data collection, we became interested in comparing performance in the VDT with the visual object MST [51]. A total of 33/66 younger adults and 35/66 older adults completed the object MST after the VDT.

## Verbal completion task

**Material.** All stimuli were displayed on a gray background on a 21" iMac using PsychoPy [52]. All 320 words were selected from WordNet, a lexical database that organizes nouns into hierarchies of relations, with the aim of modelling the lexical knowledge of a native English speaker [53]. The word stimuli ranged in frequencies from 55 to 43,605 per million words ($M = 1555$, $SD = 3985$), as determined by $SUBTL_{WF}$ (word frequency per million words; [54], were 5–10 letters long, and tended to be more abstract, ranging in concreteness scores from 1.19–5.00 ($M = 2.96$, $SD = 0.98$), as determined by concreteness ratings obtained from a norming study [55]. We chose words that were more abstract to increase the likelihood of finding close synonyms for the VDT and to reduce the dual-coding of verbal content with mental imagery [56] because the vividness of this process declines with age [57–60]. Additionally, all words had 3-letter stems that were unique across the whole stimulus set, which served as cues during the retrieval portion of the VCT. We binned the 320 words into 8 sets of 40 words and counterbalanced bin assignment to the VCT Inclusion and Exclusion, as well as the VDT tasks across participants.

**Encoding phase.** The VCT included two encoding blocks, each consisting of 20 target words with an additional two buffers at the beginning and two at the end to reduce primacy and recency effects. During encoding, a list of capitalized target words was presented to the participants, one at a time (see Fig 1B). All participants were informed that their memory of these words would be tested later. Each word was presented in the context of an easy-to-read sentence (Flesch reading ease > 60.0; Flesch-Kincaid grade level < 7) [61] to orient participants to a specific meaning of that word (e.g., "Time is a crucial ELEMENT of this plan"). Participants read each sentence out loud with the experimenter in the room to ensure they complied with processing instructions and to improve memory formation [62]. This sentence-based encoding task was deliberately chosen to provide a modest level of semantic support for all participants. Our aim was to reduce age-related variability in spontaneous strategy use – particularly older adults' documented difficulties with self-initiated encoding [63] – while minimizing executive demands. After reading each sentence, they then rated how well they knew the target word (1 = "don't know," 2 = "kind of know," and 3 = "know very well") by pressing the appropriate keyboard button. Participants were given 15 seconds to respond with an inter-stimulus interval (ISI) of 1000 ms, but trials progressed upon a response if made before the time limit.

**Retrieval phase.** We used the PDP to disentangle the automatic reactivation of past experiences enabled by pattern completion from executive processes used to guide strategic retrieval. This procedure parses indices of automatic and controlled processes from performance on an Inclusion task – in which participants use stem cues to recall studied words – and an Exclusion task – in which participants use stem cues to recall words that were not studied [43] (see details in Results).

Thus, after each encoding block, participants completed one Inclusion task with half of the studied words (10 words) and one Exclusion task with the other half of the studied words (10 words; see Fig 1C). Exclusion tasks always came after

Inclusion tasks to minimize the use of strategies developed in the exclusion task. In both tasks, each cue was a three-letter stem that corresponded to one of the studied words. For the Inclusion task, participants were asked to complete the cue with a word they had just studied. For the Exclusion task, participants were asked to complete the cue with a word they had not seen before in the study phase. Participants were given 15 seconds to respond orally with an ISI of 1000 ms; an experimenter typed responses, and the trial proceeded if a response was made before the maximum time limit. Key variables are described in the Results section.

### Verbal discrimination task

**Material.**  All stimuli were drawn from the same pool of stimuli as the VCT. As a reminder, because the 320 words had been binned and counterbalanced across tasks, each participant saw different sets of words in the VCT and VDT, with no overlap within participants. To create the VDT lures, two independent raters identified the three most frequent thesaurus-based synonyms [64,65] for each target word. For words with multiple meanings, we selected the most common meaning. We selected the closest synonym by calculating the Leacock and Chodorow's Normalized Path Length [66], which computes scaled semantic similarity between concepts c1 and c2 in WordNet, where *length* is the length of the shortest path between the two concepts, and *D* is the maximum depth of the taxonomy [66]:

$$simLC(c1, c2) = -log(length/2D)$$

We used the Leacock and Chodorow method because it has been shown to be highly correlated with human ratings, reflecting higher validity compared to other similarity measures [67,68].

**Encoding phase.**  The instructions and general format of the encoding task were the same for both the VCT and VDT; only the number of trials differed. The VDT included one encoding block with 80 words and an additional two buffers at the beginning and two at the end. There was also a 10-minute delay between VDT encoding and retrieval, in which participants were given a Sudoku puzzle. The purpose of this delay was to increase the difficulty of the task, as verbal memory recognition tests may be more prone to ceiling effects than cued recall tests, especially in younger adults [69].

**Retrieval phase.**  We modelled the design of the VDT after the 40-item, old/new recognition memory version of the MST, a task commonly used to evaluate the efficacy of individuals' pattern separation processes [15]. Participants were shown a randomized mix of 40 studied words (targets), 40 new words that were close synonyms to the other 40 studied words (lures), and 40 new unrelated words (foils). They indicated whether each word was old ('b') or new ('n'). Critically, though the lures are close synonyms to the old target words, they should be treated as new (see Fig 1D). Different item types were randomly intermixed during the task. Participants had 2 seconds to respond with an ISI of 500 ms; the task progressed after a response was made. Key variables are described in the Results section.

### Object mnemonic similarity task

We administered the 40-item object MST to a total of 33 younger and 35 older adults (see Table 1) from the full sample. The MST included an encoding phase in which participants were instructed to remember objects for a later memory test. During this phase, they viewed 80 color photographs of everyday objects and made indoor/outdoor judgments by indicating whether each object was commonly found indoors ('1') or outdoors ('2'). On each trial, items were presented for 2 seconds, followed by a blank screen and an ISI of 500 ms. Immediately after, participants were given a surprise recognition memory test in which they identified a random mixture of 40 old target objects, 40 similar lures, and 40 novel foils as old ('b') or new ('n') (see Fig 1E). These trial types were randomly intermixed during the task. Participants had 2.5 seconds to respond with an ISI of 500 ms; participants could make responses either when the item was on the screen or during the blank screen following it. Key variables are described in the Results section.

## Statistical approach

All statistical tests were two-tailed with a significance threshold of α = 0.05. For parametric tests (e.g., t-tests, linear mixed models, ANOVAs), assumptions of normality were tested using Shapiro-Wilk tests. Where post hoc comparisons were conducted, we applied Sidak corrections for multiple comparisons. To reduce the influence of extreme outliers, we applied the winsorize() function in R [70], which replaces values below the 5th percentile with the 5th percentile value, and values above the 95th percentile with the 95th percentile value. For binary outcome data (e.g., item accuracy in the VDT), we used generalized linear mixed-effects models with a binomial distribution. Model fit was assessed using AIC, and continuous predictors were scaled prior to analysis.

## Results

### Discrimination tasks

Our VDT was modelled after the MST, a widely used task to behaviourally test pattern separation. We aimed to assess if there were any age differences in the mnemonic discrimination of semantically similar items. Partway through data collection, we additionally administered the object MST to ensure that our sample replicated past findings [15]. We also conducted supplementary analyses to account for the wide age range in our older adult sample (see Supplementary Note 5), in which we found that any observed age group differences were not driven by age variability within groups.

**Lure discrimination results.** To determine whether there is an age difference in behavioural pattern separation, we followed previous research and calculated the discriminability of lures and foils ($d'_a$ [L,F]) which reflects the difference in the distributions of evidence for being old between lures and foils and is calculated as $d'_a = \frac{(\mu_{target} - \mu_{lure}) / \sqrt{\sigma^2_{target} - \sigma^2_{lure}}}{2}$ [71]. We used $d'_a$ rather than $d'$ because it doesn't assume that distributions of familiarity have equal variance across trial types [71], making it a more flexible measure. We excluded two younger and two older participants who were unable to discriminate between old target words and new foils ($d'_a$ below 0.5; chance level). For both the VDT and MST, we also excluded trials that had a RT < 200 ms [72,73], which resulted in an exclusion of 0.2% of trials for younger adults and 1.4% of trials for older adults on the VDT, and an exclusion of 0.02% of trials for younger adults and 0.01% of trials for older adults on the object MST. For the VDT only, we also excluded words that participants rated as unfamiliar (as "1") during the encoding phase, which resulted in an exclusion of 2% of trials for younger adults and 0.03% of trials for older adults.

We first replicated age-related declines in lure discrimination in the object MST. We identified that there was no age effect in object recognition (i.e., $d'_a$ (T,F); $t(60) = -1.3$, $p = .19$, 95%CI [−0.03, 0.007]) and that, across age groups, lures were more often endorsed as being old than foils ($F(1,132) = 334.34$, $p < .0001$). This pattern was consistent with the MST's intended design in which lures elicit higher false-alarm rates than foils, reflecting their perceptual similarity to targets. Following the analyses of Stark and colleagues [15], we computed a mixed 2 × 2 ANOVA on mnemonic discrimination scores ($d'_a$), using age group (Younger and Older) and $d'_a$ type (Target-Lure discrimination: $d'_a$ (T,L), and Lure-Foil discrimination: $d'_a$ (L,F) as factors. We found a significant main effect of $d'_a$ type ($F(1,132) = 67.55$, $p < .0001$), and a significant interaction ($F(1,132) = 30.62$, $p < .0001$; Fig 2B). Consistent with past MST findings, post-hoc t-tests (Sidak correction for multiple comparisons with a p-value threshold of .05) revealed that older adults had a higher $d'_a$ of lures compared to foils (i.e., $d'_a$ (L,F); see Table 2) compared to younger adults, reflecting how they were more likely to false alarm to lures than their younger counterparts (Fig 2A). Likewise, older adults showed a reduced ability at discriminating targets from lures (i.e., $d'_a$ (T,L); see Table 2) compared to younger adults. Therefore, in the MST, we see evidence for an age-related shift in lure discrimination, in which older adults treat lures more like targets and less like foils than younger adults (see Fig 2A).

We performed the same analyses as conducted in the MST for the VDT. Contrary to our findings in the MST, older adults exhibited a recognition deficit in the VDT (i.e., $d'_a$ (T,F); $t(126) = -4.5$, $p < .0001$, 95%CI [−0.06, −0.02]), and across samples, false alarm rates to lures and foils were not significantly different ($F(1,132) = 3.122$, $p = .08$). False alarm rates to foils were higher in the VDT (27% for younger adults, 39% for older adults) than in the MST (3% for younger adults, 3% for

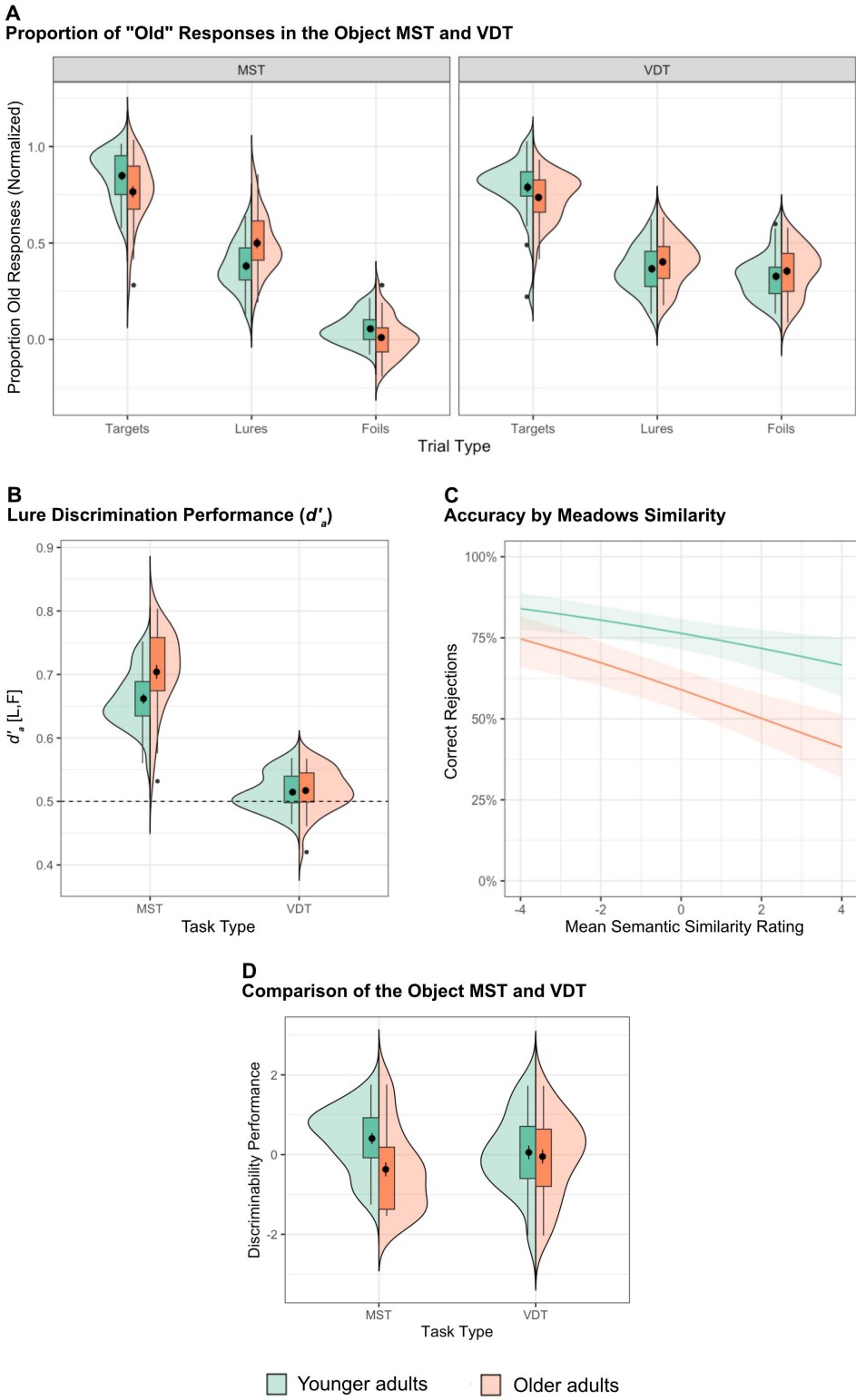

**Fig 2. A Comparison of the Object Mnemonic Similarity Task and the Verbal Discrimination Task. (A)** Proportion of old responses to target, lure, and foil trials on the Object MST and VDT. The mean proportion of old responses per condition was calculated for each participant and was normalized for within-participant error bars [74]. Notably, in the VDT, age groups did not differ in their false alarm rates to lures and foils. **(B)** Older adults have

difficulty discriminating lures compared to younger adults on the visual MST but do not significantly differ in the VDT. While $d'_a$ (L,F) is above 0.5 (chance level) in the MST, it is not significantly above chance in the VDT. In other words, while object foils solicited low false alarm rates in the MST, word foils were just as likely to be falsely endorsed as old as word lures. **(C)** Accuracy of judgments to lures and foils by normalized, mean Meadows Similarity ratings. Both age groups were more likely to false alarm to words that were semantically related to studied words, but their sensitivity to these semantic relations did not differ ($\beta=0.04$, p=.195). **(D)** Direct comparison of memory specificity in Object MST and MST. The respective scores were inversed so that higher values reflected greater accuracy in distinguishing between similar stimuli. Then, they were winsorized, checked for normality, and z-scored across all participants. There was a significant interaction between age group and task type ($\beta=0.17$, p=.047). In panels A, B, D, and all subsequent Figures, large black dots represent the group mean, the box depicts the interquartile range, and the whiskers extend to 1.5x of the interquartile range. In panel C, the lines are estimates from the mixed model (see Supplementary Methods).

**Table 2. Age Comparisons for Performance on the MST, VDT, and VCT.**

| Task | Measure | Younger Group Mean (SD) [a] | Older Group Mean (SD) | P-value |
|---|---|---|---|---|
| MST[b] | Proportion Old – Target | 0.82(0.13) | 0.79(0.21) | 0.56 |
| | Proportion Old – Lure | 0.35(0.14) | 0.53(0.21) | **0.0001** |
| | Proportion Old – Foil | 0.03(0.04) | 0.03(0.05) | 0.23 |
| | $d'_a$ (T,F) | 0.78(0.04) | 0.77(0.06) | 0.19 |
| | $d'_a$ (T,L) | 0.64(0.06) | 0.57(0.05) | **<0.0001** |
| | $d'_a$ (L,F) | 0.66(0.04) | 0.70(0.06) | **0.002** |
| VDT[c] | Proportion Old – Target | 0.75(0.16) | 0.75(0.18) | 0.88 |
| | Proportion Old – Lure | 0.30(0.16) | 0.43(0.19) | **<0.0001** |
| | Proportion Old – Foil | 0.27(0.15) | 0.39(0.20) | **0.0001** |
| | $d'_a$ (T,F) | 0.67(0.05) | 0.63(0.06) | **<0.0001** |
| | $d'_a$ (T,L) | 0.66(0.06) | 0.61(0.06) | **<0.0001** |
| | $d'_a$ (L,F) | 0.51(0.03) | 0.51(0.04) | 0.86 |
| | Slope | −0.13(0.32) | −0.18(0.38) | 0.45 |
| VCT[d] | Inclusion (proportion answered with a studied word) | 0.67(0.17) | 0.48(0.16) | **<0.0001** |
| | Exclusion (proportion answered with a studied word) | 0.23(0.14) | 0.25(0.13) | 0.46 |
| | Automatic Estimate | 0.41(0.18) | 0.31(0.14) | **0.0008** |
| | Controlled Estimate | 0.44(0.24) | 0.24(0.22) | **<0.0001** |

[a]SD = standard deviation.

[b]MST = Mnemonic Similarity Task (Object).

[c]VDT = Verbal Discrimination Task.

[d]VCT = Verbal Completion Task.

[e]All reported p-values are two-tailed.

older adults), suggesting that the novel foils may have been too similar to the studied items in the design. Nevertheless, we conducted the same mixed 2x2 ANOVA on mnemonic discrimination scores ($d'_a$), using age group (Younger and Older) and $d'_a$ type (Target-Lure discrimination: $d'_a$ (T,L), and Lure-Foil discrimination: $d'_a$ (L,F) as factors. We found significant main effects of age group ($F(1,252) = 14.4$, $p = .0002$) and $d'_a$ type ($F(1,252) = 438.23$, $p < .0001$), as well as a significant interaction ($F(1,252) = 15.75$, $p < .0001$). Post-hoc t-tests (Sidak correction for multiple comparisons with a p-value threshold of .05) revealed that older adults were worse at discriminating targets from lures (i.e., $d'_a$ (T,L)) than younger adults were ($t(252) = −5.5$, 95%CI [−0.06, −0.03], $p < .00001$), indicating that older adults were more likely to falsely identify lure items as studied targets (Fig 2A). However, there were no age differences in old responses to lures compared to foils (i.e., $d'_a$ (L,F): $t(252) = 0.122$, 95%CI [−0.015, 0.017], p=.90), consistent with the idea that memory specificity is retained with age in the semantic domain [23,24]. This conclusion would be premature, however, because behavioral performance suggested that our semantic similarity manipulation was less effective than the object similarity in the MST. Specifically,

false alarm rates to semantic foils were higher than those in the MST and closer to lure false alarms in the VDT, reducing our ability to measure the impact of semantic relatedness on false alarm rates and resulting in an outcome akin to a floor effect.

**Semantic similarity results.** The analyses above suggest that the foils in the VDT may have had unintended semantic relationships to studied words. To address this and resolve the issue of the $d'_a$ floor effect, we leveraged the high variability in false alarm rates across lures and foils to assess more rigorously how semantic similarity influences mnemonic discrimination across age groups. Specifically, we quantified how similar each new word (regardless of whether we intended it to be a lure or foil) was to each of the words studied by that participant. We assessed how various similarity metrics predicted correct rejection rates on the VDT. We reasoned that, if semantic similarity does create memory confusion, a similarity metric would be negatively related to accuracy, with shallower slopes indicating memory representations that are robust to semantic overlap and steeper slopes indicating poorer semantic discriminability. Notably, this parametric analysis of mnemonic discrimination aligns well with past empirical work in the visual domain [14,75] and computational models of pattern separation in the hippocampus [7,25,75].

Our candidate metrics of semantic similarity included model-derived and human ratings. We used two model-derived semantic similarity metrics: the Leacock and Chodorow's Normalized Path Length from WordNet (described above) and Word2Vec similarity values. Specifically, with Word2Vec, we used Wikipedia articles as a corpus to represent our word stimuli as vectors, such that words that commonly occur in similar verbal contexts are positioned close in a semantic space [76]. We also attained human ratings from a separate, online sample using a spatial multi-arrangement task [77,78] (see Supplementary Methods). For each similarity metric, we obtained two measures: the mean similarity of the studied words to each tested foil/lure, as well as the maximum similarity of the studied words to each tested foil/lure. We then entered each into its own generalized linear mixed model via the lme4 package (v1.1.35.1) [79] to assess how it predicted trial-level foil/lure accuracy (coded as correct=1, incorrect=0) (see Fig 2C). These models used a binomial distribution with a logit link, appropriate for binary outcomes. Each model also included age-group and its interaction with similarity, as well as standardized Shipley-II vocabulary scores and their interaction with similarity, word length, word concreteness/abstractness, and word frequency as covariates. We also modelled random intercepts and slopes for similarity, word length, word concreteness/abstractness, and word frequency, grouped by participant. The model that explained the most variance in participants' correct rejection rates operationalized similarity as the mean human-rated similarity of the foil/lure to all studied words (determined with AIC; see Supplementary Table 2). Accordingly, we used this definition of similarity for all subsequent analyses.

Within this model, there was a significant effect of similarity ($\beta = -0.13$, $SE = 0.03$, $p < .0001$), as well as an age group difference in correct rejection rates ($\beta = 0.33$, $SE = 0.08$, $p < .0001$). However, semantic similarity did not differentially impact older and younger adults' ability to reject lures/foils (Interaction $\beta = 0.04$, $SE = 0.03$, $p = .20$). So, while participants in both age groups were more likely to false alarm to words that were semantically similar to studied words, age was not related to this tendency. These results suggest that age-related losses in perceptual memory specificity do not carry over to the semantic domain, contrasting with the domain-general nature of pattern separation.

To this end, we were interested in directly comparing age-related differences in mnemonic discrimination across the Object MST ($d'_a$ for lure vs. foil trials) and the VDT (slope). Shapiro-Wilk tests confirmed that both metric distributions did not deviate from normal distributions (MST $d'_a$ (L,F): $W = 0.985$, $p = 0.613$; VDT slope: $W = 0.973$, $p = 0.158$). We then winsorized the data to reduce the influence of extreme values (i.e., values below the 5th percentile and above the 95th percentile were set to those respective percentile values) and normalized the scores within task across all participants so that the metrics had the same scale. After scaling, these measures of memory discriminability served as the dependent variable in a multiple linear regression model, which examined how they were predicted by the interaction between age and task type. The results revealed a significant interaction between age and task type ($\beta = 0.17$, $t(130) = 2.01$, $p = .045$). This suggests that the effect of age on performance depends on the task type, where greater declines emerge on the Object MST (Fig 2D).

One important consideration in our VDT findings is that our younger and older samples differed not only in their age but also in their language background. While 76% of older adults' first language was English (n = 50), only 44% of younger adults reported English as being their first language (n = 29). Within those whose first language is English (n = 79), we observed a trending interaction between age group and mean similarity (β = −0.17, p = .058; see Fig 3). Perhaps individuals whose first language is not English (predominately younger adults in the full sample) have more difficulty with fine-grained semantic discrimination, obscuring an age effect within the greater sample, despite having comparable vocabulary scores on the Shipley-II to native speakers. If semantic similarity were to differentially impact older adults, it would provide some support for a domain-general pattern separation account, with the caveat that memory precision for perceptual details is more influenced by age than semantic details. However, with an underpowered trend, more research that accounts for language backgrounds is needed.

### Verbal completion task

We used an independent task to tax pattern completion, indexing it as the estimate of automatic memory recall derived from the PDP. According to the PDP, correct responding in the inclusion task (intentionally completing the stem with a previously studied item) could be accomplished through controlled recollection (C), automatic memory reactivation (A), or

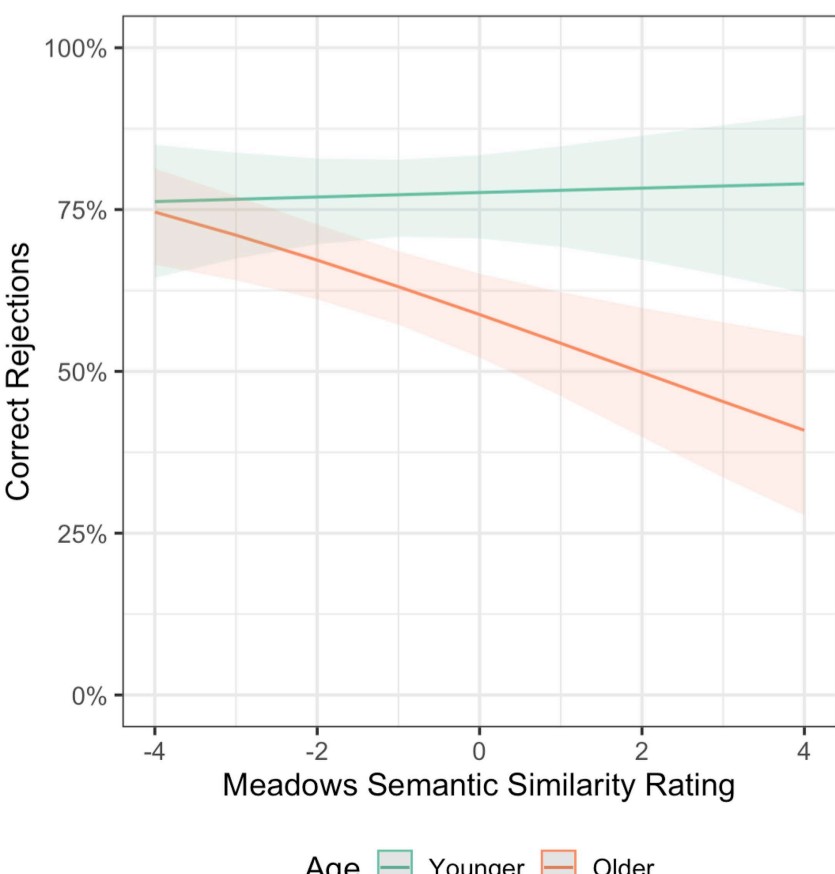

**Fig 3. Effect of Age on Performance on the Verbal Discrimination Task in Native English Speakers.** Plotted is the accuracy of judgments to lures and foils by normalized, mean Meadows Similarity ratings. Within the sample of individuals who reported English as their first language, there was a trending interaction between age and similarity, contrasting from the null effects observed in the greater sample.

both. In the exclusion task, in which participants need to complete the stem with any word other than a previously studied one, controlled recollection and automatic processes act in opposition. Specifically, a studied word can be mistakenly given if the automatic process is not opposed by cognitive control. Using linear algebra, we isolated automatic processes (A) from controlled processes (C; See Supplemental Methods). We then conducted a t-test to see if the two groups differed on these metrics, with a particular focus on the automatic estimate, which served as our index of pattern completion.

Older adults were less likely to recall the correct studied item on the Inclusion task compared to younger adults ($t$(130) = −6.48, $p$ < .0001, $d$ = 1.12), but there were no reliable age differences for responding with a previously studied item in the Exclusion tasks ($t$(130) = 0.74, $p$ = .46, $d$ = 0.13). Older adults had reduced controlled processing ($t$(130) = −5.49, $p$ < .0001, $d$ = 0.96), which was expected as aging is strongly associated with a loss of executive control [80,81]. Importantly, we found that automatic processes were also reduced in older adults ($t$(130) = −3.42, $p$ = .0008, $d$ = 0.60); this effect remained significant when accounting for Shipley-II performance in a linear regression model (β = 0.08, $SE$ = 0.02, $p$ = .02; see Fig 4), and did not depend on participants' English language backgrounds (see Supplementary Note 1). These age-related decreases in automatic memory retrieval align with some past findings [82,83], although the literature on PDP automatic processes and aging is relatively mixed (see Supplementary Table 5). In an exploratory analysis, we also investigated if this reduction in automatic estimates observed in older adults may be affected by the number of associates that compete with reactivation of the studied word at test–as would be expected if they struggled with inhibition rather than pattern completion. Specifically, we examined whether the number of possible stem completions moderated age differences in automatic reactivation. However, we did not see a significant difference across ages (β = −0.1, $SE$ = 0.05, $p$ = .06; Supplementary Note 2). Together, the observed age-related decline in automatic retrieval goes against the idea that there is a bias towards pattern completion in aging. Further, we conducted supplementary analyses including age as a continuous covariate within each group (see Supplementary Notes 4–5) and reported summary statistics of the pairwise correlations among predictor variables (Supplementary Note 6). We found that any observed age group differences were not driven by age variability within groups, and that multicollinearity is unlikely to have affected our regression analyses.

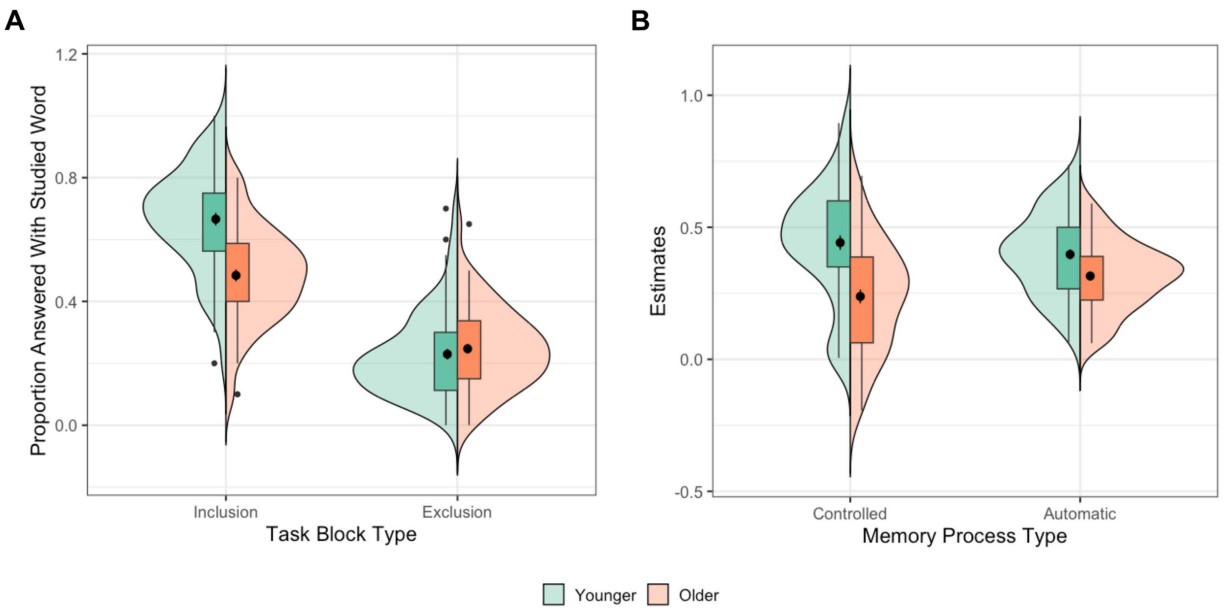

**Fig 4. Effect of Age on Performance on the Verbal Completion Task. (A)** Older adults were less likely to respond with the correct studied word in the Inclusion task. There were no reliable age differences for responding with a previously studied item in the Exclusion task. **(B)** Older adults had reduced controlled and automatic estimates derived from the PDP compared to younger adults.

**Are behavioral expressions of pattern separation and pattern completion inversely related?.** Lastly, we asked if our independent measurements of verbal behavioural pattern separation and completion were inversely related to each other, as would be predicted by the proposal that they are opposing operations [9]. To derive individual slopes reflecting semantic pattern separation, we modelled each participant's correct rejection rates with a logistic regression that included the same word covariates as were used in the full mixed model. We then partially correlated the resulting slopes with each participant's automatic retrieval score, including standardized Shipley-II scores as a covariate (see Fig 5). In contrast to the common assumption that behavioural pattern separation and completion are negatively related, they were not significantly correlated across all ages ($r(128) = -.05$, $p = .55$), nor within younger ($r(63) = .01$, $p = .91$) or older ($r(63) = .14$, $p = .27$) adults (see Supplementary Note 3 for correlations with VCT controlled estimates).

## Discussion

A reduction in memory specificity often accompanies aging. However, it is unclear whether this results from a domain-general, age-related bias from hippocampal pattern separation to pattern completion. To investigate this, we devised two verbal memory assessments and used the object MST to examine signatures of memory differences across ages. First, we tested whether older adults showed a domain-general decrease in memory specificity across semantic and perceptual domains. We found that older and younger adults similarly discriminated between semantically, but not perceptually, similar lures [23,24]. Despite the $d'_a$ analysis being limited by floor effects, the overall pattern of findings was nonetheless consistent with the results of the slope analysis, which was not subject to the same limitation. Interestingly, though, younger adults had marginally greater memory specificity in the semantic domain than older adults when we accounted for language backgrounds, warranting future research within this domain. Second, we found that older adults are less likely to engage in automatic reactivation processes than younger adults, inconsistent with them having a bias toward pattern completion. Overall, our results challenge the view that aging is uniformly associated with deficits in memory specificity across modalities and that there is an ensuing bias towards pattern completion. These findings are discussed in greater detail below.

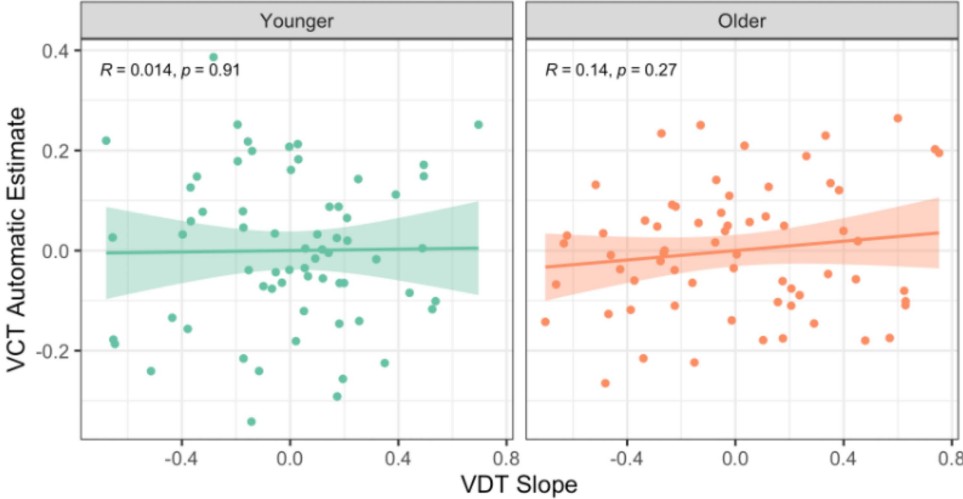

**Fig 5. Correlation of Performance on the Verbal Discrimination and Completion Tasks.** VCT = Verbal Completion Task; VDT = Verbal Discrimination Task. There were no significant correlations between automatic estimates from the VCT and similarity slopes from the VDT in either younger or older adults. This suggests that our independently derived estimates of behavioural pattern completion and separation were not inversely related to each other.

In our study, we leveraged the predictions of the pattern separation account to inform the debate on the nature of reduced memory specificity in aging. While the pattern separation account predicts a domain-general decline in pattern separation, our results align with previous work suggesting that age-related decline in memory specificity may be more evident in perceptual domains than in semantic ones [23,24,84]. Importantly, we extend previous demonstrations using narratives and related words to precisely measure the semantic similarity of all new words at test to study, thus providing a nuanced and objective measure of semantic discrimination in aging. These findings suggest a shift in how memory specificity changes with age, which may reflect both a compensatory strategy for reduced episodic fidelity and a broader change in cognitive priorities. As older adults experience declines in detail-specific memory, they may also place greater value on meaningful, conceptually relevant information. This motivational shift may enhance semantic processing and neural specificity (hyperdifferentiation) for conceptual information [85–88], potentially benefitting performance in tasks like the VDT, which require fine-grained discrimination of conceptually similar words.

However, this apparent preservation in semantic discrimination with age may also be related to inherent differences in the representational complexity across domains. Perceptual stimuli often contain more nuanced, detail-rich features (e.g., color, spatial resolution, viewer perspective) than conceptual stimuli, and this complexity may place a greater burden on memory fidelity, making it more susceptible to age-related decline. Therefore, the observed interaction between age and stimulus type may reflect these differences in complexity, rather than the comparative preservation of conceptual details in memory. Indeed, alternative accounts of age-related memory decline, such as the 'cluttered memory' hypothesis [89], suggest that older adults may retain more irrelevant features, which can interfere with tasks requiring high representational fidelity. From this perspective, semantic representations, having fewer details, may be less susceptible to such interference. Domain differences in complexity are also relevant for gist-based accounts of aging, in which older adults are argued to retain the general features of an experience, rather than its exact details [90–92]. These types of representations may be more suited to semantic discrimination than more detail-sensitive perceptual discrimination. Thus, the dissociation we observe between perceptual and semantic memory decline may not reflect a fundamental domain-based distinction, but rather differences in the complexity of the stimuli themselves. This challenge of disentangling perceptual and conceptual contributions is a central issue for psychological research; addressing it will inspire major methodological and theoretical innovations moving forward.

Beyond the varied interpretations of preserved semantic discrimination, our VDT results also must be interpreted with caution. Alike to how visual processing abilities may contribute to performance on visual mnemonic discrimination tasks [93], individuals' language processing – beyond vocabulary level and crystallized knowledge – may affect performance on semantic mnemonic discrimination tasks. In our exploratory analyses, we assessed whether language history influenced the findings because of demographic differences between our younger and older adult samples. Particularly, our sample of younger adults was mainly from the University of Toronto, which has a highly international population, and only 44% reported English as their first language despite being enrolled in university programs taught in English and having comparable Shipley-II scores as native speakers. On the other hand, our older adults were recruited from the greater community, and 76% reported English as their first language. In the native English-speaking sub-sample, a trend toward age-related impairment emerged, with younger adults more accurately rejecting lures across similarity levels than older adults, who showed reduced discrimination of fine-grained semantic details. Note that this trend aligns with the findings of Ilyés, Paulik, and Kereztes [22], a study conducted in Budapest, Hungary, where the language landscape may be more homogeneous compared to Toronto and other similar multicultural cities. If older adults have heightened semantic knowledge, why do they not show superior semantic discrimination? Perhaps the trending decline we observed in the native English-speaking sample may reflect age-related changes that are at odds with each other: a domain-general decline in hippocampal pattern separation and hyperdifferentiation of semantic details in neocortical regions. Future neuroimaging work could provide further clarity.

 

Our study also revealed the complexity of manipulating semantic relations. Our first attempt – using close synonyms of individual studied items – followed a similar rationale to that used in the MST development but did not yield the expected effect. Participants found many of the words that we designed to be foils (synonyms) just as familiar as the words that we designed to be lures. Had we stopped there, we could have concluded that both younger and older participants did not confuse semantically related words in our memory task. Instead, we systematically characterized all semantic relations within our word lists to reveal a robust impact of semantic similarity across our samples. In doing so, we also demonstrated how ratings from humans were more related to memory performance than similarity estimates from popular word embedding models (see Supplementary Table 2). It may be that WordNet and Word2Vec similarity metrics miss features that are important for how people mentally represent word meaning – either because of corpus limitations (e.g., the corpus in Ilyés, Paulik & Kereztes [22] was limited to the press corpus) or because these models emphasize formal or distributional similarity. In contrast, human similarity ratings may reflect more experiential, functional, or affective associations between words, which could make them better predictors of memory specificity. These human-rated norms and stimulus sets are publicly available on OSF to encourage more memory researchers to consider the semantic space in which memory items are embedded. Future researchers may also consider manipulating semantic relationships by altering conceptual similarity in visual stimuli [94,95] or by rearranging word pairs with high and low similarity, as demonstrated by Ilyés et al [22].

To index behavioural pattern completion, we focused on the automatic reactivation of past experiences. While older adults showed relatively preserved memory specificity in the semantic domain (VDT), we observed marked deficits in their ability to automatically reactivate previously studied items, regardless of language backgrounds. Within a relatively mixed PDP literature, our results support past studies demonstrating declines in automatic processing with age [82,83] though these deficits may be more pronounced in experiments that require less generation and elaboration during encoding (see Supplementary Note 5). Our findings also argue against age-related biases toward pattern completion. To bolster this conclusion, we explored the counterintuitive possibility that older adults may have lower automatic retrieval scores precisely because they have over-active pattern completion. Specifically, they might complete stems with the recently studied word less often because they have difficulties inhibiting their recall of other words related to the stem [96]. However, older adults were not more likely to struggle when stems had many potential associates – if anything, retrieval interference was more pronounced in younger adults. Overall, our findings add to demonstrations that pattern completion is impaired in aging [37,38]. However, without a direct measure of memory reactivation [35], it remains possible that older adults were biased to pattern complete irrelevant memories. It is also worth noting that, while verbal in nature, the three letter cues in the VCT could be considered more perceptual than semantic. Accordingly, age-related bias toward semantic details may have been less helpful in the VCT compared to the VDT.

Our results should also be considered within the context of some limitations. One is our use of the PDP: while it is helpful in dissociating more effortful controlled processes from automatic ones, the procedure assumes independence between the two processes, which may not always be the case [97,98]. However, since our study does not compare controlled processes to automatic processes, the lack of independence of the two processes is less of a concern. A major limitation of this study is the groups' difference in linguistic diversity, and, thus, the demographic differences between the younger and older adult samples. Yet, this group difference helped illuminate the importance of linguistic background and language history on verbal memory, warranting attention in future research. Additionally, our choice of encoding task –sentence reading – could have influenced our findings. We intentionally designed it to support consistent encoding strategies across groups, but it is possible that the conceptual processing that it promoted could have disproportionately benefitted older adults, who tend to rely on semantic and schematic support. Lastly, another important consideration is our reliance on single tasks to estimate pattern separation and pattern completion. Ideally, convergent validity across multiple independent tasks measuring the same constructs would strengthen confidence in these interpretations. Achieving this would require an experimental design that systematically includes multiple complementary tasks to provide convergent evidence

for each cognitive process. While this was beyond the scope of the present study, we acknowledge this as an important direction for future research to enhance construct validity and better characterize the mechanisms underlying age-related memory changes.

In future studies, neuroimaging data could further connect these behavioral findings to neural representations in the hippocampus and neocortex. Our focus on refining behavioral assessments offers a useful complement to prior neuroimaging work, and we hope it inspires future research investigating how the aging brain encodes and automatically retrieves semantic details.

On a broader note, our findings seemingly contrast with a rich literature linking mnemonic discrimination deficits to changes within the hippocampal network [9,32,51]. However, it is possible, if not likely, that both reduced hippocampal pattern separation and changes in neocortical representations [86,99,100] contribute to the observed behaviour. Neural dedifferentiation is a phenomenon where neural distinctiveness of posterior cortical representations is reduced with age [101,102]. In certain scenarios, declines in both may make the problem additive: if dedifferentiated perceptual representations enter the hippocampus and must then be orthogonalized by a declining dentate gyrus network, then catastrophic interference could occur due to one or both factors. As observed in prior research, a minimum threshold of discriminability must be achieved before pattern separation can be effective [103]. In other cases, these changes may offset each other: if hyperdifferentiated semantic representations enter the aging hippocampus, additional pattern separation may not be needed to avoid interference, and behavioural performance may be preserved. What, then, are the implications for pattern completion? First, recalling a task-relevant memory could be limited if poorly differentiated neocortical representations were encoded (with compounding effects for hippocampal pattern separation). At the time of retrieval, superfluous neocortical signals could also degrade retrieval cues or derail the hippocampally-driven pattern completion in the neocortex. And, if hippocampal pattern completion is over-active, irrelevant memories could be more often completed, interfering with the reactivation of the correct representation. Further complicating matters is that in most tasks, declining executive functions could exacerbate the problem, for example, by impairing the source monitoring required to determine if the reactivated memory is contextually appropriate. However, if the cue-memory relationship emphasizes semantic features, hyperdifferentiation in this domain may overcome the hurdles. With all these potential threats to successfully recalling a probed memory, it is not surprising that we observed such steep age-related declines in the VCT. An important caveat to our argument, though, is that while these domain distinctions may be clearer in the lab, it is rare for information to be precisely just semantic or solely perceptual in the real world. Yet, this multidimensionality of experience could also be leveraged by older adults to work towards their strengths in the semantic domain.

## Conclusion

Recent research shows mixed findings in lure discrimination abilities within semantic but not perceptual domains [22–24]. Our work adds to this growing literature by also finding that older adults' discrimination performance in the semantic, verbal domain is relatively preserved, even when accounting for vocabulary level. Further, we found that older adults exhibit reduced behavioural pattern completion compared to younger adults. Together, our findings provide evidence that challenges the domain-general account of pattern separation and lends insight to changes in our goal states: as we age, we may tend to focus less on fine-grained perceptual information, and instead, place greater importance on shared meaning. Ultimately, our work helps unravel some of the complexity in how the representation of details in memory changes with age.

## Supporting information

**S1 File. Contains Supplementary Methods and Supplementary Notes.**
(DOCX)

**S1 Fig. Effect of age on performance on the verbal completion task in Native English Speakers.** Within the sample of individuals who reported English as their first language, effects in the VCT remained the same as the full sample. Older adults had reduced controlled and automatic estimates compared to younger adults.
(TIFF)

**S2 Fig. Effect of the number of common associates on automatic estimates for the VCT.** Individuals who studied words with a greater number of common stem-completion associates had more difficulty automatically reactivating the precise studied word at test ($p < 0.0001$), and this effect did not significantly differ across age groups.
(TIFF)

**S3 Fig. Correlation between VCT controlled estimates and VDT slopes.** VCT = Verbal Completion Task; VDT = Verbal Discrimination Task. There were no significant correlations between controlled estimates from the VCT and similarity slopes from the VDT in either younger or older adults.
(TIFF)

**S4 Fig. Correlation between Age and VCT automatic.** VCT = Verbal Completion Task. There were no significant correlations between automatic estimates from the VCT and age with the full older adult sample.
(TIFF)

## Acknowledgments

The authors would like to thank the following individuals for their dedication to data collection: Emily Williams, Polina Rybitska, Ella Ma, Marta Shahezian, and Rekha Ravikumar. The authors would also like to thank Jasper van den Bosch and Christopher Chow for helpful discussions and insights on conducting similarity matrix analyses.

## Author contributions

**Conceptualization:** Ariana Youm, Melanie Cohn, Katherine Duncan.

**Data curation:** Ariana Youm.

**Formal analysis:** Ariana Youm.

**Funding acquisition:** Ariana Youm, Melanie Cohn, Katherine Duncan.

**Investigation:** Ariana Youm, Melanie Cohn, Katherine Duncan.

**Methodology:** Ariana Youm, Melanie Cohn, Katherine Duncan.

**Project administration:** Ariana Youm.

**Resources:** Melanie Cohn, Katherine Duncan.

**Supervision:** Melanie Cohn, Katherine Duncan.

**Validation:** Ariana Youm.

**Visualization:** Ariana Youm.

**Writing – original draft:** Ariana Youm.

**Writing – review & editing:** Ariana Youm, Melanie Cohn, Katherine Duncan.

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
