## [Decision Letter · Decision Letter 0]

20 Jun 2025

Dear Dr. Youm,

Thank you for submitting your manuscript to PLOS ONE. After careful consideration, we feel that it has merit but does not fully meet PLOS ONE’s publication criteria as it currently stands. Therefore, we invite you to submit a revised version of the manuscript that addresses the points raised during the review process.

We look forward to receiving your revised manuscript.

Kind regards,

Jie Wang, Ph.D.

Academic Editor

PLOS ONE

Journal Requirements: 

4. We notice that your supplementary figures are uploaded with the file type 'Figure'. Please amend the file type to 'Supporting Information'. Please ensure that each Supporting Information file has a legend listed in the manuscript after the references list.

Additional Editor Comments:

The authors should address the comments raised by the three reviewers.

Reviewers' comments:

Reviewer's Responses to Questions

**Comments to the Author**

1. Is the manuscript technically sound, and do the data support the conclusions?

Reviewer #1: Partly

Reviewer #2: Yes

Reviewer #3: Yes

2. Has the statistical analysis been performed appropriately and rigorously?

Reviewer #1: Yes

Reviewer #2: Yes

Reviewer #3: Yes

3. Have the authors made all data underlying the findings in their manuscript fully available?

Reviewer #1: Yes

Reviewer #2: Yes

Reviewer #3: Yes

4. Is the manuscript presented in an intelligible fashion and written in standard English?

Reviewer #1: Yes

Reviewer #2: Yes

Reviewer #3: Yes

Reviewer #1: The authors did a good job of incorporating prior research, theory, and methodology into their approach. The manuscript was well written, and the figures worked well to display the findings. The authors examined whether pattern completion and/or pattern separation are impaired in older adults versus younger adults using several complimentary tasks: the Mnemonic Similarity Task (visual), and two novel tasks: Verbal Discrimination Task (verbal) and the Verbal Completion Task (verbal). As expected, based on prior similar studies using the MST, they found that older adults exhibited impaired “Old”-Lure proportions, Target vs Lure d’, and Lure vs Foil d’, but intact “Old”-Target proportions, “Old”-Foil proportions, and Target vs Foil d’. Importantly, there was a significant interaction between age group and d’ type (Target-Lure d’ vs. Lure-Foil d’). This indicated that the older adults were more likely to treat lures like targets (and less like foils) than the younger adults. For the VDT, they found impaired “Old”-Lure proportions, “Old”-Foil proportions, Target vs Foil d’, and Target vs Lure d’, but intact “Old”-Target proportion and Lure vs Foil d’. Importantly, there was no impairment in Lure vs Foil d’, suggesting that memory specificity was intact in the semantic domain. However, neither younger nor older adults were able to discriminate lures from foils in the VDT (floor effect), making it difficult to determine whether there was an age-related impairment in memory specificity. Direct comparisons across groups between the Lure vs Foil d’ for the MST and VDT revealed more age-related differences for the MST than the VDT. Follow-up analyses of the effect of language familiarity or semantic-similarity on performance did not appear to be relevant the group comparisons of memory specificity in the semantic domain. For VCT, they found impaired Inclusion proportion as well as impaired estimates of automatic and controlled retrieval, whereas the Exclusion proportion was intact. There was no significant correlation identified between the VDT slope (thought to reflect pattern separation) and the automatic estimate from the VCT (thought to reflect pattern completion).

Concerns:

1. The authors found that older adults exhibited different levels of memory specificity on the MST than the VDT (i.e., lure vs foil d’) and stated “This suggests that the effect of age on performance depends on the task type, where greater declines emerge on the Object MST”. This conclusion is difficult to support given the floor effects for the Foil vs Lure d’ for both groups on the VDT. If both groups were above chance on the VDT and the same significant interaction emerged, this would be strong evidence for their claim. It appears they have created a task in which it is impossible for humans to discriminate between lures and foils, which is the critical type of discrimination needed to justify the claim they are trying to make about memory specificity being differentially impaired across tasks in older adults, i.e., that memory specificity is not a domain-general type of impairment that occurs during aging.

2. For the results supporting the analyses in Figure 5, the authors note there was no significant relationship between independent estimates of pattern completion from the VCT and pattern separation from the VDT. Yet, it is possible that these two estimates are uncorrelated because they are from two different tasks. It would be stronger evidence for their claim if they could show that independent estimates of pattern separation from both tasks were significantly correlated and/or that independent estimates of pattern completion from both tasks were significantly correlated. Against that backdrop, their finding that pattern separation and pattern completion were uncorrelated would be more convincing.

Minor Concerns:

1. Abstract: Words like “change” and “decline” are used, but given the non-longitudinal, group-based comparisons these types of words are not accurate. The results should be described in terms of deficits or impairment.

2. Page 17, line 343: “confirming desirable features of the MST design in our sample”. What desirable features are the authors referring to?

3. Page 19, line 389: “This conclusion would be premature however, because our manipulation of semantic similarity was not as pronounced as the object similarity manipulation in the MST”. Please make it clearer how the authors are making the determination that semantic similarity was not as pronounced as on the object similarity in the MST. It is not clear what characteristics or findings they are looking at to make that decision.

4. Page 29, line 610. The authors state that a limitation of their behavioral study was that it did not include neuroimaging data. Reporting on behavior alone within a conceptual framework, but without accompanying neuroimaging data, is not a limitation.

Reviewer #2: ## Summary

The present article, "Aging and episodic memory specificity: Evidence challenging a domain-general pattern separation decline" explores (1) the loss of specificity in healthy aging, found in visual episodic memory, for conceptual similarity and (2) how pattern separation and pattern completion are interdependent for this situation. The study is based on three experimental tasks, a classic mnemonic similarity task, an adapted version manipulating semantic similarity and a verbal completion task to assess semantic pattern completion.

The topic is timely and relevant. Increasing data support the loss of specificity for visual episodic memory. Yet, little is known for conceptual material. In the same vein, it is supposed that pattern completion is interdependent with pattern separation, but behavioral results remain mixed.

Overall, the article is well written, based on an original and adapted methodology, and proposes an interesting contribution to the topic. The introduction offers a comprehensive review of the literature. The experimental paradigms are well designed. The statistical analyses are correctly done and the conclusion is consistent with the data.

Nonetheless, I will have relatively major comments (see below) that should be addressed leading to a major revision.

## Major Comments

One of the study’s aims, stated in the introduction section, is “3) We accounted for other cognitive processes, such as executive functioning or visual-spatial processing”, but no mention of the objective is made in the abstract. I suggest adding this information there.

I am also confused about the control mentioned. The authors used a PDP (process dissociation procedure) procedure, which is suitable to account for the attentional and controlled process, but it is unclear how the visual-spatial processing is taken into account. No further mention of visuo-spatial processing is done in the article. This mention should be removed or relevant data should be added.

Moreover, it is stated that “we administered a neuropsychological test battery to older adults only”, but that these data will not be discussed here. Nonetheless, executive and visuo-spatial results from the neuropsychological assessment should be used to account for the third objective of the paper. By the way, please add, at least as supplementary data, the list of the tests used.

On a statistical level, the age range is very large for the older adults considering that aging decline accelerated around the 70’s. The analysis method should control for it. In the same idea, the education level of the older adults is very high and significantly different from the one in the young adults group. This variable should also be controlled.

I would recommend exploring the correlation matrix between all variables and/or to use, at least ANCOVA, or better, hierarchical regressions or generalized linear mixed as used after (by the way, a yes/no recognition task is more like a Bernoulli distribution than a normal distribution which shouldn’t be processed by ANOVA, see Dixon, 2008).

Some statistical details are also missing such as the use of bilateral or unilateral p-values tests, the control of outliers for accuracy and reaction times, other assumptions checks (normality), and so on.

Please comment on the possible bias related to intentional memorizing content in the semantic adaptation of the MST vs. the incidental learning done in the standard MST.

Perceptual information is richer than conceptual ones. As analogous signals, perceptual information are filled with details (e.g. color nuances, resolution, context, points of view, and so on) that are not presented in a verbal concept (there is a multiple of possible representations of an ELEMENT). Consistent with this idea, phonological similarity induces a similar loss of specificity. Consequently, it is difficult to conclude that the interaction between age and the nature of the stimuli is supported by a preserved processing of the conceptual information. One could imagine that the effect is mainly driven by the degree of details convey in the stimuli used (see also the sentence l. 390 “semantic similarity was not as pronounced as the object similarity manipulation in the MST”). Could the authors better justify their conclusions or acknowledge this hypothesis as a limit of their study. Otherwise, please add more nuanced to the discussion (e.g. “our results add to emerging evidence that age-related decline in memory specificity is restricted to perceptual domains...” ).

## Minor Comments

- All the data should be uploaded on the OSF as dropbox is a suitable and durable sharing method (the data remain on the folders of someone specifically and the access could be easily removed).

- The pictures should be uploaded with higher quality (e.g. 600 DPI).

- Update the abstract to mention the aim to add the classical MST task for a subset of participants (i.e. direct comparison of visual vs semantic similarity, see l).

- For the same reason, update the introduction section presenting the aims of the study (l-139) to explicitly present the rational to “We additionally included the standard object MST for comparison with the VDT for a subset of participants”. If it is an objective of the study, it should be presented with the other ones (l. 125-131) and if it a secondary or exploratory aims, it should be mentioned as it (see l. 210 “Midway through data collection, we became interested in comparing performance in the VDT with the visual object MST.”).

- Please add more details about the power analysis. Was it computed for a simple t-test/ANOVA test or did it account for the hierarchical levels or possible covariables.

- Please provide more details on the ethical approval (e.g. approval id or number).

- In general, please add Notes to the tables to provide details on the acronyms and abbreviations (SD, F, M, MST...). Please also add where the group comparisons were significant.

- I would not agree with the following statement (l. 206-208) “this way, it is more likely that any observed age-related memory impairments are driven by mnemonic changes rather than visual acuity or perceptual changes.”. The pattern separation literature is heavily based on representational-hierarchical approach which did not consider a difference in nature between memory and perceptual processes (see Kent et al., 2016; see also embodied view of cognition Mille et al., 2021).

- It is odd to mix up separated at a similar level the verbal stimuli and the tasks using them. I suggest using a hierarchical structure with something like “Semantic tasks” at a higher level and within it “Verbal stimuli”, “Intentional Encoding Task”.

- L. 321-325 : the paragraph is mainly a redundant part of the method, please remove it.

- Please provide the exact formula used to compute the discriminability index (l. 329).

- Letters used in mathematical expressions should be in italic (F, t, p...) like in F(1,132) = 333.34, p < .

- Please change the verb “tended” (l. 386) as it could imply that the effect is not really significant.

- Please remove this statement for the results “semantic similarity was not as pronounced as the object similarity manipulation in the MST.” or provide more explanations about it, here.

- Please use consistent terminology between semantic and domain-general terms (e.g. l. 522).

- What the authors mean by "reflect a lack of interest in perceptual details” (l. 539-540)?

- I would recommend being more nuanced about “a robust deficit in automatically recall memories” (l. 584) as numerous studies have reported the opposite and as the tasks constraints should be taken into account (e.g. trigram completion has been reported as being more demanding than a real automatic memory recall and to involve some executive components).

## References

- Dixon, P. (2008). Models of accuracy in repeated-measures designs. *Journal of Memory and Language, 59*, 447–456.

- Kent, B. A., Hvoslef-Eide, M., Saksida, L. M., & Bussey, T. J. (2016). The representational–hierarchical view of pattern separation : Not just hippocampus, not just space, not just memory? _Neurobiology of Learning and Memory_, _129_, 99‑106.

- Mille, J., Brambati, S. M., Izaute, M., & Vallet, G. T. (2021). Low-Resolution Neurocognitive Aging and Cognition : An Embodied Perspective. _Frontiers in Systems Neuroscience_, _15_, 687393.

Reviewer #3: This study investigates the impact of semantic relatedness on age differences in pattern separation and completion. The authors designed two new tasks, namely the verbal discrimination and completions tasks. A subset of participants also completed the standard version of the mnemonic similarity task (MST). The results showed that the age-related decline in memory specificity is stronger for perceptual than for semantic relatedness. Older adults were also impaired in pattern completion. The authors interpret these results as evidence against the view that typical aging is associated with a domain-general decline in pattern-separation abilities and a bias toward pattern-completion.

The study question is relevant and topical. Also, the effort the authors have made to design the new tasks, and to characterize the semantic relatedness of all list words, are remarkable. The results of this study will be informative for the cognitive aging scientific community. However, there are some limitations that need to be addressed, which I list below.

1. The design of the present study includes some specific features that must be considered. First, the encoding procedure used for the VCT and VDT is likely to have oriented participants toward a deeper conceptual processing of the target words. Given that numerous previous studies have established that older adults benefit from semantic (and schematic) support, such an encoding strategy may have been especially beneficial to this group compared with younger adults, tempering age differences. Unless the authors disagree with this view, it should be acknowledged in the manuscript. Second, encoding was explicit in the VCT and VDT but incident in the MST; could the authors explain this methodological choice, and how this is considered in the interpretation of the results? Finally, from my understanding, there was no delay between encoding and test in the VCT whereas there was a 10-minute delay in the VDT – is that correct? If so, could the authors also explain this choice and its potential influence on the results?

2. I believe that the authors could further discuss the implications of their results for the nature of memory representations in typical aging. For example, the concept of “gist” is insufficiently developed. The authors mention that older adults tend to rely more on gist memory (p. 3, lines 55-56), but do not define gists. Some relevant references could be cited in the introduction and discussion, such as Grilli & Sheldon (2022), and/or Greene & Naveh-Benjamin (2023), and/or Brainerd & Reyna (2015). Another hypothesis is that memory representations in healthy older adults are “cluttered”, that is, they include more irrelevant and no-longer relevant features, in addition to relevant features, which harms performance in memory tasks that require memory representations of high fidelity (Amer et al., 2023).

Grilli, M. D., & Sheldon, S. (2022). Autobiographical event memory and aging : Older adults get the gist. Trends in Cognitive Sciences, 26(12), 1079 1089. https://doi.org/10.1016/j.tics.2022.09.007

Greene, N. R., & Naveh-Benjamin, M. (2023). Adult age-related changes in the specificity of episodic memory representations : A review and theoretical framework. Psychology and Aging, 38(2), 67 86. https://doi.org/10.1037/pag0000724

Brainerd, C. J., & Reyna, V. F. (2015). Fuzzy-trace theory and lifespan cognitive development. Developmental Review, 38, 89 121. https://doi.org/10.1016/j.dr.2015.07.006

Amer, T., Wynn, J. S., & Hasher, L. (2022). Cluttered memory representations shape cognition in old age. Trends in Cognitive Sciences, 26(3), 255 267. https://doi.org/10.1016/j.tics.2021.12.002

3.A. The observation of a trend for age x similarity interaction effect on correct rejections in the VDT in the participants who reported English as first language is important. Figure 3 distinctly shows that in young adults there was no relation between semantic similarity and correct rejections, whereas these two variables are negatively associated in older adults. Considering that this analysis was conducted in a subsample, it is likely to be underpowered. Thus, if this effect existed, it would go against the authors’ conclusions, for example “our results add to emerging evidence that age-related decline in memory specificity is restricted to perceptual domains, compared to semantic ones” (p. 25, lines 535-537), or “In contrast to older adults’ mainly preserved memory specificity in the semantic domain” (p. 27, line 580). Unless the authors can argue against this, I believe that these interpretations should be nuanced.

3.B. Related to this, the authors highlight the question of why older adults “do not show superior semantic discrimination” although they have “heightened semantic knowledge” (lines 561-562, p.26). It is important to keep in mind that the tasks designed in this study and those used in most similar studies (including that of Ilyés, Paulik, and Keresztes, 2024, SciRep), rely on episodic memory, although the conceptual similarity of targets and lures can be manipulated. Therefore, performance on these tasks is impacted by the fidelity of episodic memory representations, which declines with typical aging. This could partially explain why older adults’ performance on such tests is impaired although they have enhanced semantic knowledge. What do the authors think about this tentative interpretation?

4. Can the authors specify explicitly on what data the multiple linear regression model that aimed at comparing age differences between the object MST and VDT was conducted (p.21, lines 433-442)? My question aims to confirm that the dependent variable here corresponds to memory discrimination, since the authors report an age x task type interaction. If the dependent variable did not correspond to a mnemonic discrimination measure, then a three-way interaction (i.e., age x task type x stimulus, e.g., lure/foil) would be warranted to ensure that the effect can be interpreted as reflecting memory specificity rather than overall memory performance. Line 436 seems to indicate that, but making it explicit could be helpful for future readers.

5. As highlighted by the authors p.27, the use of verbal materials and synonyms to manipulate semantic relations between stimuli is relevant but also comes with limitations. Mentioning ways to overcome such limitations in future studies could be helpful to readers. For example, some researchers have manipulated conceptual similarity for visual material (e.g., Frick et al, 2023; Naspi et al, 2021). Alternatively, the material used by Ilyés et al., 2024 also seems to overcome this limitation, as the related/unrelated lure manipulation comes from the rearrangement of word pairs (e.g., “police uniform” and “official uniform”).

Frick, A., Besson, G., Salmon, E., & Delhaye, E. (2023). Perirhinal cortex is associated with fine-grained discrimination of conceptually confusable objects in Alzheimer’s disease. Neurobiology of Aging, 130, 1‑11. https://doi.org/10.1016/j.neurobiolaging.2023.06.003

Naspi, L., Hoffman, P., Devereux, B., Thejll-Madsen, T., Doumas, L. A. A., & Morcom, A. (2021). Multiple dimensions of semantic and perceptual similarity contribute to mnemonic discrimination for pictures. Journal of Experimental Psychology. Learning, Memory, and Cognition, 47(12), 1903‑1923. https://doi.org/10.1037/xlm0001032

Minor comments

• I am not sure that the study by Delarazan et al. (2023, Learning and memory) cited p.4 (lines 92-94) can be described as investigating “perceptual and semantic” discrimination. Rather, the authors tested memory for narrative details, that is, “moments described” (i.e., how the story unfolded). While I acknowledge that this is close to the semantic domain, it must not be confused with manipulations of conceptual similarity such as the one in the present study or that of Ilyés et al., 2024.

• P.10, line 204: “PC and PS” these abbreviations were not defined in the article.

• P.9, lines 130-131 or lines 136-137, I suggest citing also the study of Gellersen et al. (2021) to support the authors’ argument.

Gellersen, H. M., Trelle, A. N., Henson, R. N., & Simons, J. S. (2021). Executive function and high ambiguity perceptual discrimination contribute to individual differences in mnemonic discrimination in older adults. Cognition, 209, 104556. https://doi.org/10.1016/j.cognition.2020.104556

• p.8, lines 181-182, is the 95% CI of 4.77-6.96 the CI around the t-test? If so, it is surprising that this interval does not include the t value itself (i.e., 10.59). Can the authors double check and correct if needed?

• p.23, lines 481-482 and lines 489-490 seem to repeat the same information (i.e., that VCT performance was not influenced by English language background).

• P.27, line 581, it is unclear what “in their ability to recall memories” refers to, especially since there was no proper memory recall task in this study. Also, the sentence lines 583-584 is almost identical to the first sentence of this paragraph (lines 580-581).

• p.11, line 248, there seems to be a typo in “Based off of these cues”.

**Do you want your identity to be public for this peer review?** For information about this choice, including consent withdrawal, please see our Privacy Policy

Reviewer #1: No

Reviewer #2: **Yes: ** Guillaume T Vallet

Reviewer #3: No

---

## [Author Response · Author response to Decision Letter 1]

15 Aug 2025

Response to Reviewers

Author Note: Dear Reviewers, we appreciate the time and effort you have devoted to evaluating our manuscript. We are grateful for the thoughtful comments and constructive suggestions, which have significantly contributed to improving the clarity and quality of our work. Please find attached a detailed, point-by-point response to each comment, with our responses provided in blue. All page and line references refer to the revised manuscript with tracked changes. As the tracked changes formatting differs across Word (.doc) files and PDF, we have provided both the Word doc page and line number as well as the PDF page and line number for your reference.

Reviewer 1

Summary: The authors did a good job of incorporating prior research, theory, and methodology into their approach. The manuscript was well written, and the figures worked well to display the findings. The authors examined whether pattern completion and/or pattern separation are impaired in older adults versus younger adults using several complimentary tasks: the Mnemonic Similarity Task (visual), and two novel tasks: Verbal Discrimination Task (verbal) and the Verbal Completion Task (verbal). As expected, based on prior similar studies using the MST, they found that older adults exhibited impaired “Old”-Lure proportions, Target vs Lure d’, and Lure vs Foil d’, but intact “Old”-Target proportions, “Old”-Foil proportions, and Target vs Foil d’. Importantly, there was a significant interaction between age group and d’ type (Target-Lure d’ vs. Lure-Foil d’). This indicated that the older adults were more likely to treat lures like targets (and less like foils) than the younger adults. For the VDT, they found impaired “Old”-Lure proportions, “Old”-Foil proportions, Target vs Foil d’, and Target vs Lure d’, but intact “Old”-Target proportion and Lure vs Foil d’. Importantly, there was no impairment in Lure vs Foil d’, suggesting that memory specificity was intact in the semantic domain. However, neither younger nor older adults were able to discriminate lures from foils in the VDT (floor effect), making it difficult to determine whether there was an age-related impairment in memory specificity. Direct comparisons across groups between the Lure vs Foil d’ for the MST and VDT revealed more age-related differences for the MST than the VDT. Follow-up analyses of the effect of language familiarity or semantic-similarity on performance did not appear to be relevant the group comparisons of memory specificity in the semantic domain. For VCT, they found impaired Inclusion proportion as well as impaired estimates of automatic and controlled retrieval, whereas the Exclusion proportion was intact. There was no significant correlation identified between the VDT slope (thought to reflect pattern separation) and the automatic estimate from the VCT (thought to reflect pattern completion).

Author response: Thank you for your thoughtful and constructive comments. We appreciate your recognition of our integration of prior research, theory, and methodology, as well as your positive remarks about the writing and the clarity of the figures. We are happy to hear that the overall structure and presentation of the study were effective in conveying the key findings. Your feedback directly contributed to strengthening both the clarity and precision of the manuscript.

Concerns:

1. The authors found that older adults exhibited different levels of memory specificity on the MST than the VDT (i.e., lure vs foil d’) and stated “This suggests that the effect of age on performance depends on the task type, where greater declines emerge on the Object MST”. This conclusion is difficult to support given the floor effects for the Foil vs Lure d’ for both groups on the VDT. If both groups were above chance on the VDT and the same significant interaction emerged, this would be strong evidence for their claim. It appears they have created a task in which it is impossible for humans to discriminate between lures and foils, which is the critical type of discrimination needed to justify the claim they are trying to make about memory specificity being differentially impaired across tasks in older adults, i.e., that memory specificity is not a domain-general type of impairment that occurs during aging.

Author response: Thank you for raising this thoughtful and important comment. Specifically, here you raise the concern that the different effects of age on memory discrimination across modalities could reflect a floor effect since we failed to solicit significant foil-lure discrimination in our VDT design.

We agree that our primary conclusions depend on a robust impact of similarity on memory in both the MST and VDT. Indeed, this is why we cautioned the reader that, “false alarm rates to foils were higher in the VDT (27% for younger adults, 39% for older adults) than in the MST (3% for younger adults, 3% for older adults), suggesting that the novel foils may have been too similar to the studied items in the design” (Word doc page 21 lines 578-581; PDF page 23 lines 481-484). In our original design, we did not consider the degree of similarity of each stimuli (including the foils) to the rest of the stimuli. Thus, we revised the manuscript to state that any conclusions using d’ “would be premature, however, because behavioral performance suggested that our semantic similarity manipulation was less effective than the object similarity in the MST; specifically, false alarm rates to semantic foils were higher than those in the MST and closer to lure false alarms in the VDT, reducing our ability to measure the impact of semantic relatedness on false alarm rates” (Word doc page 21 lines 592-596; PDF page 24 lines 496-501).

To overcome this hurdle, we completed a comprehensive series of semantic similarity analyses which compared the similarity of each new word, regardless of our initial lure vs. foil designation, to the full set of studied words. This regression analysis revealed a robust relation between semantic similarity and false alarm rates (β=-0.13, SE=0.03, p<.0001). We used the slope of this relationship in each participant as an alternative measurement of how semantic similarity influenced memory discriminability. So, by shifting our analytic approach, we achieved your recommendation that both groups show above chance discrimination on the VDT. Critically, it’s this statically robust slope that we entered the cited interaction analysis–not the insignificant lure vs. foil d’.

We hope our explanation clarifies that we agree with your reasoning and that we prioritized identifying a robust influence of semantic similarity on memory before moving forward with modality comparisons.

We also carefully re-read our results and discussion sections to identify aspects of our writing that may have led to the confusion over which metric was used in the critical interaction analysis. To make it clearer, we added a clarification on Word doc page 23 in lines 674-675; PDF page 27 lines 558-560 saying: “To this end, we were interested in directly comparing age-related difference in mnemonic discrimination across the Object MST (d’a for lure vs. foil trials) and the VDT (slope)”.

2. For the results supporting the analyses in Figure 5, the authors note there was no significant relationship between independent estimates of pattern completion from the VCT and pattern separation from the VDT. Yet, it is possible that these two estimates are uncorrelated because they are from two different tasks. It would be stronger evidence for their claim if they could show that independent estimates of pattern separation from both tasks were significantly correlated and/or that independent estimates of pattern completion from both tasks were significantly correlated. Against that backdrop, their finding that pattern separation and pattern completion were uncorrelated would be more convincing.

Author response: We were not exactly clear what was being asked here, but we understood your suggestion to be that stronger evidence for the lack of correlation between pattern separation (PS) and pattern completion (PC) would come from demonstrating converging estimates of each construct across tasks – for example, obtaining independent estimates of PS from both the VDT and VCT (or PC from both), and then correlating those. We interpreted your logic to be that if same-construct estimates converge across tasks, this would bolster confidence in our interpretation that PS and PC are separable processes, especially if, against that backdrop, the lack of correlation between PS and PC holds.

We agree that this would be a strong approach if the tasks in question provided independent estimates of the same construct. However, in our design, the VCT was explicitly intended to measure PC, and the VDT was designed to measure PS. While it is technically possible to derive a PS-like estimate from the VCT (e.g., by inverting the PC estimate), or a PC-like estimate from the VDT, such measures would not be independent. They would be inherently mathematically coupled. For example, the PS estimate from the VCT would simply be the inverse of the PC estimate, meaning any resulting correlation (or lack thereof) would be uninterpretable. This artificial dependency would violate the assumptions needed to meaningfully assess whether PS and PC are dissociable.

Thus, while we agree in principle that cross-task convergent validity would strengthen the argument, it would require a different experimental design in which multiple tasks provide independent estimates of each construct. This is an excellent direction for future work, and we now acknowledge this more explicitly in the revised manuscript: “another important consideration is our reliance on single tasks to estimate pattern separation and pattern completion. Ideally, convergent validity across multiple independent tasks measuring the same constructs would strengthen confidence in these interpretations. Achieving this would require an experimental design that systematically includes multiple complementary tasks to provide convergent evidence for each cognitive process. While this was beyond the scope of the present study, we acknowledge this as an important direction for future research to enhance construct validity and better characterize the mechanisms underlying age-related memory changes” (Word doc page 32 lines 1006-1023; PDF page 39 lines 833-840).

Minor Concerns:

1. Abstract: Words like “change” and “decline” are used, but given the non-longitudinal, group-based comparisons these types of words are not accurate. The results should be described in terms of deficits or impairment.

Author response: Thank you for pointing this out. We have replaced them with “deficits” and “differences” instead, as seen on Word doc page 2 lines 45- 46; PDF page 2 lines 46-48.

2. Page 17, line 343: “confirming desirable features of the MST design in our sample”. What desirable features are the authors referring to?

Author response: We clarified this with: “This pattern was consistent with the MST’s intended design in which lures elicit higher false-alarm rates than foils, reflecting their perceptual similarity to targets”, on Word doc page 19 lines 531-533; PDF page 22 lines 443-445.

3. Page 19, line 389: “This conclusion would be premature however, because our manipulation of semantic similarity was not as pronounced as the object similarity manipulation in the MST”. Please make it clearer how the authors are making the determination that semantic similarity was not as pronounced as on the object similarity in the MST. It is not clear what characteristics or findings they are looking at to make that decision.

Author response: Thank you for pointing this out. We have changed it to: “This conclusion would be premature, however, because behavioral performance suggested that our semantic similarity manipulation was less effective than the object similarity in the MST; specifically, false alarm rates to semantic foils were higher than those in the MST and closer to lure false alarms in the VDT, reducing our ability to measure the impact of semantic relatedness on false alarm rates.” on Word doc page 21 lines 592-596; PDF page 24 lines 496-501.

4. Page 29, line 610. The authors state that a limitation of their behavioral study was that it did not include neuroimaging data. Reporting on behavior alone within a conceptual framework, but without accompanying neuroimaging data, is not a limitation.

Author response: Thank you for this suggestion. We agree, and we have reframed the section as a future direction: “In future studies, neuroimaging data could further connect these behavioral findings to neural representations in the hippocampus and neocortex. Our focus on refining behavioral assessments offers a useful complement to prior neuroimaging work, and we hope it inspires future research investigating how the aging brain encodes and automatically retrieves semantic details.”( Word doc page 33 lines 1024-1028; PDF page 39 lines 841-845).

Reviewer 2

Summary: The present article, "Aging and episodic memory specificity: Evidence challenging a domain-general pattern separation decline" explores (1) the loss of specificity in healthy aging, found in visual episodic memory, for conceptual similarity and (2) how pattern separation and pattern completion are interdependent for this situation. The study is based on three experimental tasks, a classic mnemonic similarity task, an adapted version manipulating semantic similarity and a verbal completion task to assess semantic pattern completion.

The topic is timely and relevant. Increasing data support the loss of specificity for visual episodic memory. Yet, little is known for conceptual material. In the same vein, it is supposed that pattern completion is interdependent with pattern separation, but behavioral results remain mixed.

Overall, the article is well written, based on an original and adapted methodology, and proposes an interesting contribution to the topic. The introduction offers a comprehensive review of the literature. The experimental paradigms are well designed. The statistical analyses are correctly done and the conclusion is consistent with the data.

Nonetheless, I will have relatively major comments (see below) that should be addressed leading to a major revision.

Author response: Thank you for your thoughtful review. We sincerely appreciate your recognition of the originality of our methodological approach and the clarity of the manuscript. We are especially grateful for your remarks on the rigor of the experimental design and statistical analyses, as well as your acknowledgment that this work adds meaningfully to a complex and evolving literature. Your insights were both encouraging and helpful as we revised the manuscript.

## Major Comments

One of the study’s aims, stated in the introduction section, is “3) We accounted for other cognitive processes, such as executive functioning or visual-spatial processing”, but no mention of the objective is made in the abstract. I suggest adding this information there.

Author response: Thank you for your suggestion. We have modified the abstract to include these details: “The current study aimed to investigate whether aging affects memory discrimination for semantically similar content, using tasks that minimize reliance on visual-spatial processing and executive functioning, both of which tend to decline with age.” on Word doc page 2 lines 34-37; PDF page 2 lines 34-37.

I am also confused about the control mentioned. The authors used a PDP (process dissociation procedure) procedure, which is suitable to account for the attentional and controlled process, but it is unclear how the visual-spatial processing is taken into account. No further mention of visuo-spatial processing is done in the article. This mention should be removed or relevant data should be added.

Author response: Thank you for pointing this out. We acknowledge that our original phrasing “accounting for” visual-spatial processing may have been misleading. To clarify, we employed verbal tasks that do not rely on visuo-spatial processing. We have revised the manuscript accordingly to make this clearer: “We accounted for executive functioning and used tasks that were less reli

---

## [Decision Letter · Decision Letter 1]

16 Sep 2025

Dear Dr. Youm,

Thank you for submitting your manuscript to PLOS ONE. After careful consideration, we feel that it has merit but does not fully meet PLOS ONE’s publication criteria as it currently stands. Therefore, we invite you to submit a revised version of the manuscript that addresses the points raised during the review process.

We look forward to receiving your revised manuscript.

Kind regards,

Jie Wang, Ph.D.

Academic Editor

PLOS ONE

Journal Requirements:

Additional Editor Comments :

The authors need to address the comments raised by Reviewer 1.

Reviewers' comments:

Reviewer's Responses to Questions

**Comments to the Author**

Reviewer #1: (No Response)

Reviewer #2: All comments have been addressed

Reviewer #3: All comments have been addressed

2. Is the manuscript technically sound, and do the data support the conclusions?

Reviewer #1: Yes

Reviewer #2: Yes

Reviewer #3: Yes

3. Has the statistical analysis been performed appropriately and rigorously?

Reviewer #1: Yes

Reviewer #2: Yes

Reviewer #3: Yes

4. Have the authors made all data underlying the findings in their manuscript fully available?

Reviewer #1: Yes

Reviewer #2: Yes

Reviewer #3: Yes

5. Is the manuscript presented in an intelligible fashion and written in standard English?

Reviewer #1: Yes

Reviewer #2: Yes

Reviewer #3: Yes

Reviewer #1: The authors have addressed my primary concern about d’ floor effects in the VDT task by explaining that the analysis of VDT slope gets around the floor effects. In addition, the lack of a group effect in the slope analysis is consistent with the lack of a group effect in the d’ analysis. I missed this logic in the initial submission, and it is still missing from the revision. Thus, as a naïve reader of the revision, I would still have the same concern about floor effects that I had upon reading the previous version.

As it stands, the semantic similarity analysis of slopes section begins “The analyses above suggest that the foils in the VDT may have had unintended semantic relationships to studied words. To address this,...”

This introduction to the semantic similarity analysis section is still missing the important issue that has arisen in the prior paragraph, where the floor effect d’ analyses are reported. The authors should make it clearer at this stage of reporting that the slope analysis will address the floor effect issue they have just read about. Also make it clearer in the Discussion that the findings from the murky d’ analysis that suffers from floor effects is nonetheless concordant with the findings from the slope analysis that does not suffer from floor effects.

Finally, given the importance of the slopes for this paper, please report the slope descriptive statistics and p values in Table 2.

All other concerns have been addressed.

Reviewer #2: The authors have done an excellent job to take into account all the comments. The responses and modifications done lead to a valuable article.

Reviewer #3: I commend the authors for the thorough revision of their manuscript. All my comments have been addressed satisfactorily.

**Do you want your identity to be public for this peer review?** For information about this choice, including consent withdrawal, please see our Privacy Policy

Reviewer #1: No

Reviewer #2: **Yes: ** Guillaume T Vallet

Reviewer #3: No

---

## [Author Response · Author response to Decision Letter 2]

16 Oct 2025

Response to Reviewers

Reviewer 1

Reviewer #1: The authors have addressed my primary concern about d’ floor effects in the VDT task by explaining that the analysis of VDT slope gets around the floor effects. In addition, the lack of a group effect in the slope analysis is consistent with the lack of a group effect in the d’ analysis. I missed this logic in the initial submission, and it is still missing from the revision. Thus, as a naïve reader of the revision, I would still have the same concern about floor effects that I had upon reading the previous version.

As it stands, the semantic similarity analysis of slopes section begins “The analyses above suggest that the foils in the VDT may have had unintended semantic relationships to studied words. To address this,...”

This introduction to the semantic similarity analysis section is still missing the important issue that has arisen in the prior paragraph, where the floor effect d’ analyses are reported. The authors should make it clearer at this stage of reporting that the slope analysis will address the floor effect issue they have just read about.

Author: Following your suggestion, we have modified our section that now reads as: “This conclusion would be premature, however, because behavioral performance suggested that our semantic similarity manipulation was less effective than the object similarity in the MST. Specifically, false alarm rates to semantic foils were higher than those in the MST and closer to lure false alarms in the VDT, reducing our ability to measure the impact of semantic relatedness on false alarm rates and resulting in an outcome akin to a floor effect.

Semantic similarity results

The analyses above suggest that the foils in the VDT may have had unintended semantic relationships to studied words. To address this and resolve the issue of the d′a floor effect, we leveraged the high variability in false alarm rates across lures and foils to assess more rigorously how semantic similarity influences mnemonic discrimination across age groups.”

Reviewer: Also make it clearer in the Discussion that the findings from the murky d’ analysis that suffers from floor effects is nonetheless concordant with the findings from the slope analysis that does not suffer from floor effects.

Author: Thank you for your suggestion. We have now added this line to our Discussion: “Despite the d′a analysis being limited by floor effects, the overall pattern of findings was nonetheless consistent with the results of the slope analysis, which was not subject to the same limitation.”

Reviewer: Finally, given the importance of the slopes for this paper, please report the slope descriptive statistics and p values in Table 2.

Author: Thank you for pointing this out. We added the values to Table 2.

Reviewer: All other concerns have been addressed.

Author: Thank you for your thoughtful points. We really appreciated all of your comments!

---

## [Decision Letter · Decision Letter 2]

20 Oct 2025

Aging and episodic memory specificity: Evidence challenging a domain-general pattern separation decline

PONE-D-25-21591R2

Dear Dr. Youm,

We’re pleased to inform you that your manuscript has been judged scientifically suitable for publication and will be formally accepted for publication once it meets all outstanding technical requirements.

Kind regards,

Jie Wang, Ph.D.

Academic Editor

PLOS ONE

Additional Editor Comments (optional):

Reviewers' comments:

Reviewer's Responses to Questions

**Comments to the Author**

Reviewer #1: All comments have been addressed

2. Is the manuscript technically sound, and do the data support the conclusions?

Reviewer #1: Yes

3. Has the statistical analysis been performed appropriately and rigorously?

Reviewer #1: Yes

4. Have the authors made all data underlying the findings in their manuscript fully available?

Reviewer #1: Yes

5. Is the manuscript presented in an intelligible fashion and written in standard English?

Reviewer #1: Yes

Reviewer #1: (No Response)

**Do you want your identity to be public for this peer review?** For information about this choice, including consent withdrawal, please see our Privacy Policy

Reviewer #1: No

---

## [Editor Report · Acceptance letter]

PONE-D-25-21591R2

PLOS ONE

Dear Dr. Youm,

I'm pleased to inform you that your manuscript has been deemed suitable for publication in PLOS ONE. Congratulations! Your manuscript is now being handed over to our production team.

Kind regards,

on behalf of

Dr. Jie Wang

Academic Editor

PLOS ONE